# A selective ER-phagy exerts procollagen quality control via a Calnexin-FAM134B complex

Alison Forrester[1,†], Chiara De Leonibus[1,†], Paolo Grumati[2,†], Elisa Fasana[3,†], Marilina Piemontese[1], Leopoldo Staiano[1], Ilaria Fregno[3,4], Andrea Raimondi[5], Alessandro Marazza[3,6], Gemma Bruno[1], Maria Iavazzo[1], Daniela Intartaglia[1], Marta Seczynska[2], Eelco van Anken[7], Ivan Conte[1], Maria Antonietta De Matteis[1,8], Ivan Dikic[2,9,*] (iD), Maurizio Molinari[3,10,**] (iD) & Carmine Settembre[1,11,***] (iD)

## Abstract

Autophagy is a cytosolic quality control process that recognizes substrates through receptor-mediated mechanisms. Procollagens, the most abundant gene products in Metazoa, are synthesized in the endoplasmic reticulum (ER), and a fraction that fails to attain the native structure is cleared by autophagy. However, how autophagy selectively recognizes misfolded procollagens in the ER lumen is still unknown. We performed siRNA interference, CRISPR-Cas9 or knockout-mediated gene deletion of candidate autophagy and ER proteins in collagen producing cells. We found that the ER-resident lectin chaperone Calnexin (CANX) and the ER-phagy receptor FAM134B are required for autophagy-mediated quality control of endogenous procollagens. Mechanistically, CANX acts as co-receptor that recognizes ER luminal misfolded procollagens and interacts with the ER-phagy receptor FAM134B. In turn, FAM134B binds the autophagosome membrane-associated protein LC3 and delivers a portion of ER containing both CANX and procollagen to the lysosome for degradation. Thus, a crosstalk between the ER quality control machinery and the autophagy pathway selectively disposes of proteasome-resistant misfolded clients from the ER.

**Keywords** autophagy; Calnexin; collagen; endoplasmic reticulum; FAM134B
**Subject Categories** Autophagy & Cell Death; Membrane & Intracellular Transport
The EMBO Journal (2019) 38: e99847

## Introduction

Macroautophagy (hereafter referred to as autophagy) is a homeostatic catabolic process devoted to the sequestration of cytoplasmic material in double-membrane vesicles (autophagic vesicles, AVs) that eventually fuse with lysosomes where cargo is degraded (Mizushima, 2011). Autophagy is essential to maintain tissue homeostasis and counteracts both the onset and progression of many disease conditions, such as ageing, neurodegeneration and cancer (Levine *et al*, 2015).

Substrates can be selectively delivered to AVs through receptor-mediated processes. Autophagy receptors harbour a LC3 or GABARAP interaction motif (LIR or GIM, respectively) that facilitate binding of the cargo to LC3 or GABARAP proteins, which decorate autophagosomal membranes (Stolz *et al*, 2014; Rogov *et al*, 2017). Proteins and entire organelles or their portions can be targeted to autophagy via receptor-mediated processes. A notable example is represented by ER-phagy, a selective form of autophagy in which portions of the ER are sequestered within AVs and transported to the lysosomes for degradation (Fregno & Molinari, 2018; Grumati *et al*, 2018). To date, the yeast Atg39, Atg40 and the mammalian FAM134B, SEC62, RTN3 and CCPG1 proteins have been identified as ER-phagy receptors (i.e. as LC3-binding proteins that decorate specific ER subdomains for capture by AVs) (Khaminets *et al*, 2015; Mochida *et al*, 2015; Fumagalli *et al*, 2016; Grumati *et al*, 2017; Smith *et al*, 2018). ER-phagy mediates the

1  Telethon Institute of Genetics and Medicine (TIGEM), Pozzuoli, Italy
2  Institute of Biochemistry II, Goethe University Frankfurt – Medical Faculty, University Hospital, Frankfurt am Main, Germany
3  Faculty of Biomedical Sciences, Institute for Research in Biomedicine, Università della Svizzera italiana (USI), Bellinzona, Switzerland
4  Department of Biology, Swiss Federal Institute of Technology, Zurich, Switzerland
5  Experimental Imaging Center, San Raffaele Scientific Institute, Milan, Italy
6  Graduate School for Cellular and Biomedical Sciences, University of Bern, Bern, Switzerland
7  Division of Genetics and Cell Biology, San Raffaele Scientific Institute, Ospedale San Raffaele, Milan, Italy
8  Department of Molecular Medicine and Medical Biotechnologies, University of Naples "Federico II", Naples, Italy
9  Buchmann Institute for Molecular Life Sciences, Goethe University Frankfurt, Frankfurt am Main, Germany
10 School of Life Sciences, École Polytechnique Fédérale de Lausanne, Lausanne, Switzerland
11 Department of Medical and Translational Science, University of Naples "Federico II", Naples, Italy
   *Corresponding author. Tel: +49 69 6301 5964; E-mail: dikic@biochem2.uni-frankfurt.de
   **Corresponding author. Tel: +41 91 8200319/352; E-mail: maurizio.molinari@irb.usi.ch
   ***Corresponding author. Tel: +39 081 1923 0601; E-mail: settembre@tigem.it
   † These authors contributed equally to this work

turnover of ER membranes and promotes recovery after ER stress, bacterial and viral infections (Khaminets *et al*, 2015; Chiramel *et al*, 2016; Fumagalli *et al*, 2016; Grumati *et al*, 2017; Lennemann & Coyne, 2017; Moretti *et al*, 2017; Smith *et al*, 2018).

ER homeostasis relies on ER quality control mechanisms to prevent the accumulation of inappropriately folded cargoes within its lumen. Misfolded proteins are dislocated from the ER to the cytosol to be degraded by the 26S proteasome, a process known as ER-associated degradation (ERAD)(Preston & Brodsky, 2017). However, not all misfolded ER proteins are eligible for ERAD and thus must be cleared from the ER through other processes. Autophagy-dependent and autophagy-independent lysosomal degradation of proteins from the ER has also been reported (Ishida *et al*, 2009; Hidvegi *et al*, 2010; Houck *et al*, 2014; Fregno *et al*, 2018). However, the mechanism by which misfolded ER luminal proteins are recognized by the cytosolic autophagic machinery and delivered to the lysosomes remains to be understood.

Collagens are the most abundant proteins in animals, and type I and type II collagen (COL1 and COL2) are the major protein components of bone and cartilage, respectively (Bateman *et al*, 2009). They are synthesized as alpha I and alpha II chains and folded into triple helices of procollagen (PC) in the ER. Properly folded PCs associate with the heat shock protein 47 (HSP47) chaperone and then leave the ER through sub-regions called ER exit sites (ERES), within COPII-coated carriers, and move along the secretory pathway (Malhotra & Erlmann, 2015). Previous studies estimated that approximately 20% of newly synthesized type I PC (PC1) is degraded by lysosomes as a consequence of inefficient PC1 folding or secretion (Bienkowski *et al*, 1986; Ishida *et al*, 2009). In case of mutations in PC or HSP47, the fraction of PC degraded increases significantly (Ishida *et al*, 2009). Similarly, a fraction of type II PC (PC2) produced by chondrocytes of the growth plates is degraded by autophagy, and inactivation of this catabolic pathway results in PC2 accumulation in the ER and defective formation of the extracellular matrix (Cinque *et al*, 2015; Bartolomeo *et al*, 2017; Settembre *et al*, 2018). Overall these data clearly indicate that aberrant PC molecules represent ERAD-resistant substrates where autophagic clearance emerges as a crucial and physiologically relevant event in the maintenance of cellular and organ homeostasis. However, to date, the mechanism by which ER-localized PCs are selectively disposed of by autophagy is still unknown.

In this study, we sought to uncover the mechanisms that select non-native PC in the ER lumen for lysosomal delivery and clearance. We found that the misfolded PC molecules (e.g. HSP47 negative) are cleared from the ER through FAM134B-mediated ER-phagy. Notably, FAM134B binds PC molecules in the ER through the interaction with the transmembrane ER chaperone Calnexin (CANX) that acts as a specific FAM134B ER-phagy co-receptor for misfolded PCs. The formation of this complex allows the selective delivery of PC molecules to the lysosomes.

# Results

### Autophagy promotes degradation of intracellular procollagens preventing their accumulation in the ER

Using three different collagen producing cell lines, mouse embryonic fibroblasts (MEFs) and human osteoblasts (Saos2) stably expressing the autophagosome membrane marker LC3 fused with GFP (GFP-LC3) (Kabeya *et al*, 2000), and rat chondrosarcoma cells (RCS) immunolabelled for LC3, we observed co-localization of LC3-positive vesicles (hereafter referred as autophagic vesicles, AVs) with PC1 (MEFs and Saos2) and PC2 (RCS) (Fig 1A–D). Similarly, we observed the co-localization of PC1 spots with the GFP-tagged double-FYVE domain-containing protein 1 (DFCP1), which labels sites for autophagosome biogenesis (omegasomes) (Fig EV1A and B). *In vivo*, osteoblasts of the mandible in Medaka fish embryos (stage 40), showed the presence of AVs containing PC2 molecules (Fig EV1C–E).

When MEFs, Saos2 and RCS cells were treated with the lysosomal inhibitor bafilomycin A1 (BafA1), PC molecules accumulated in the lumen of swollen endo/lysosomes (LAMP1-positive organelles, hereafter referred as lysosomes) (Fig 1E–G). These data were validated by PC1 immuno-electron microscopy (IEM) (Fig 1H). Western blot analysis confirmed the accumulation of intracellular PCs, as well as of the autophagy markers LC3-II and SQSTM1/p62, in cells treated with BafA1 compared to untreated cells (Fig EV2A). BafA1 washout induced a rapid clearance of PC1 and PC2 from lysosomes of MEFs and RCS, respectively, in line with the notion that PCs are degraded in this compartment (Fig EV2B and C).

Lysosomal storage disorders (LSDs) are genetic diseases characterized by a defective lysosomal degradative capacity due to mutations in genes encoding for lysosomal proteins. As a result, lysosomal substrates progressively accumulate within the lumen of lysosomes causing lysosomal swelling and cell dysfunction. We sought to determine whether PC molecules accumulate in the lysosomes of LSD osteoblasts. Saos2 osteoblasts in which the alpha-L-iduronidase gene was deleted using CRISPR-Cas9 technology (CRISPR-IDUA) represent a disease model of mucopolysaccharidosis type I (MPS I), a lysosomal storage disorder with severe skeletal manifestations (Oestreich *et al*, 2015). Similar to cells treated with BafA1, CRISPR-IDUA showed swollen lysosomes, suggesting an accumulation of undigested substrates in the lysosomal lumen (Fig 1I). Most importantly, the level of PC1 in lysosomes, and in the whole cell lysate, was higher in CRISPR-IDUA Saos2 compared to control cells (Fig 1I and J).

To verify at which trafficking stage PC became an autophagy substrate, we performed a temperature shift assay where PC accumulates in the ER during incubation at 40°C, and is released from the ER upon shift of the temperature to 32°C. U2OS cells expressing GFP-LC3, mCherry-PC2 and ER marker RDEL-HALO, were imaged upon shift to 32°C (time 0 s). We observed that PC2 spots formed at the ER and progressively accumulated GFP-LC3 (Fig 2A and Movie EV1). Similarly in U2OS cells expressing phosphatidylinositol 3-phosphate (PtdIns(3)P) -recognition domain construct GFP-2•FYVE, mCherry-PC2 and ER marker RDEL-HALO, the PC2 was visible at an area of GFP-2•FYVE-positive ER, and dissociated from the main tubular ER structure releasing a vesicle positive for ER, GFP-2•FYVE and PC2 (Fig 2B and Movie EV2). Co-localization between GFP-LC3, the ER chaperone CANX and PC1 was also observed by Airyscan super-resolution confocal microscopy (Fig 2C). Similarly, we observed co-localization of PC1 spots with GFP-DFCP1 and CANX in MEFs and Saos2 cells (Fig EV3A). We also performed correlative light electron microscopy (CLEM) and electron tomography of GFP-LC3 expressing Saos2 cells, showing that PC1 and CANX are found together in a small vesicle contained within a larger LC3-positive vesicle (Fig 2D and E). Taken together, these data suggest that PC

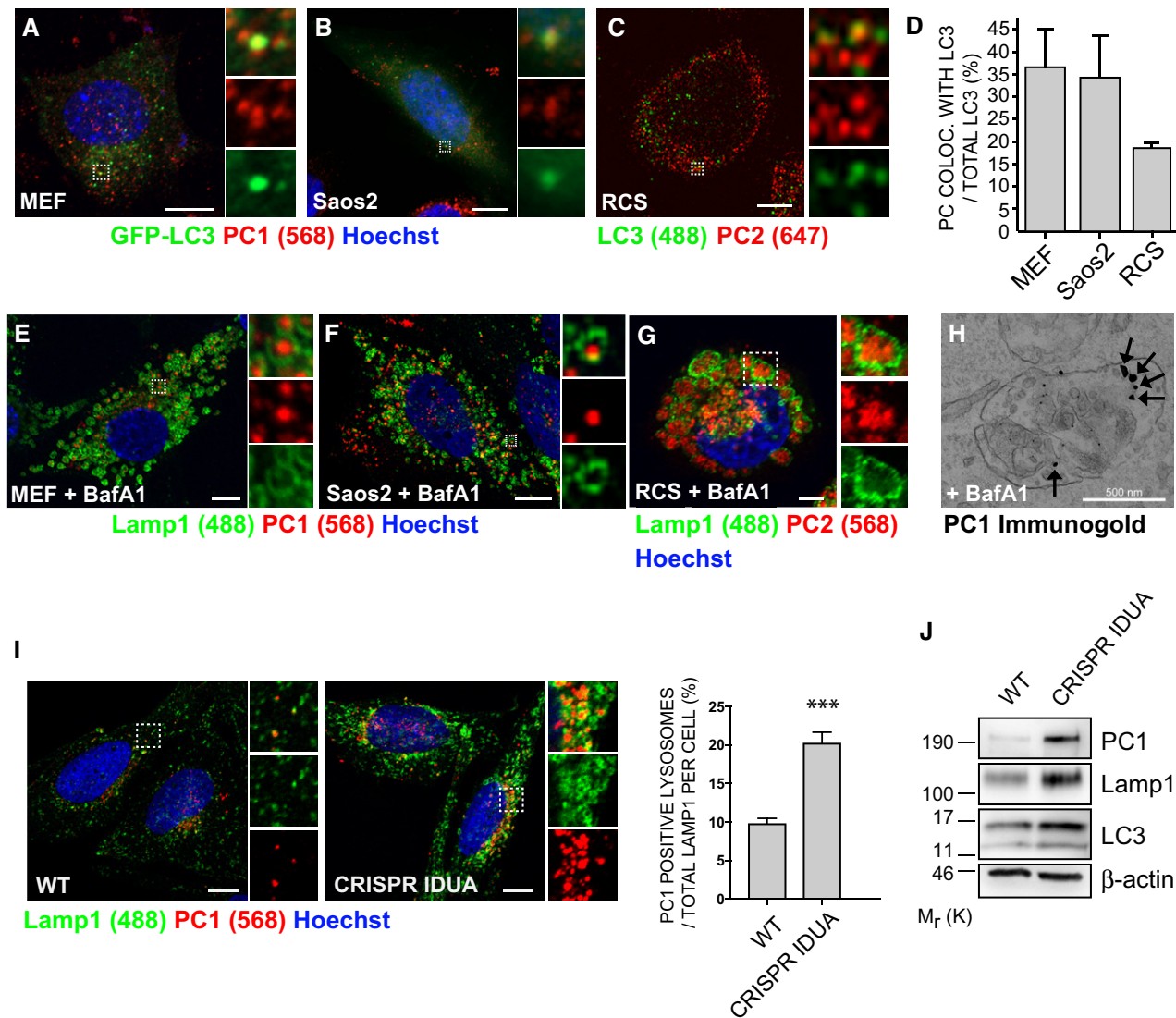

**Figure 1. PCs are autophagy substrates and accumulate in lysosomes.**

A, B  Airyscan confocal analysis of PC1 (568, red) co-localization with GFP-LC3 (green) in (A) MEF (B) Saos2. Scale bars = 10 μm. The insets show higher magnification (A = x4.68; B = x6.76) and single colour channels of the boxed area.

C  Airyscan confocal analysis of PC2 (647, red) co-localization with LC3 (488, green) in RCS cells. Scale bars = 10 μm. The insets show higher magnification (x7.33) and single colour channels of the boxed area.

D  Quantification of GFP (A, B) or LC3 (C) vesicles positive for PC1 or PC2, expressed as % of total LC3 (mean ± SEM), n = 18 cells (MEFs and Saos2); n = 12 (RCS) from three independent experiments.

E–G  Scanning confocal microscopy analysis of MEFs, Saos2 and RCS cells treated with BafA1, immunolabelled for PC1 or PC2 and LAMP1. Nuclei were stained with Hoechst. (E, F) Scale bars = 10 μm, (G) Scale bars = 5 μm. The insets show higher magnification (E = x4.99; F = x6.49; G = x2.01) and single colour channels of the boxed area.

H  Transmission EM analysis in Saos2 cells, treated with BafA1, showing in detail a lysosome which contains immunolabelled PC1 (with nanogold particles), as indicated by arrows.

I  Scanning confocal microscopy analysis of Saos2 WT and CRISPR-Cas9 *IDUA* Saos2 at steady state, immunolabelled for PC1 and LAMP1. Nuclei were stained with Hoechst. Scale bar = 10 μm. The insets show higher magnification (left = x3.09; right = x3.12) and single colour channels of the boxed area. Bar graph shows quantification of lysosomes containing PC1 expressed as % of total LAMP1 per cell (mean ± SEM). n = 31 WT cells, n = 33 CRISPR cells counted; three independent experiments. Student's unpaired, two-tailed *t*-test ***$P$ < 0.0001.

J  WT and CRISPR-*IDUA* Saos2 lysed and analysed by Western blot. Data are representative of three independent experiments.

Source data are available online for this figure.

molecules are sequestered within LC3-positive vesicles when they are still within the ER.

The collagen-specific chaperone HSP47 was excluded from the AVs containing PC1 in MEFs, strongly suggesting that autophagy sequesters non-native PC1 molecules in the ER (Fig EV3B), in line with previous results (Ishida *et al*, 2009; Cinque *et al*, 2015). To further corroborate this notion, we studied two missense mutations in the COL2A1 protein (R789C and G1152D) that induce misfolding

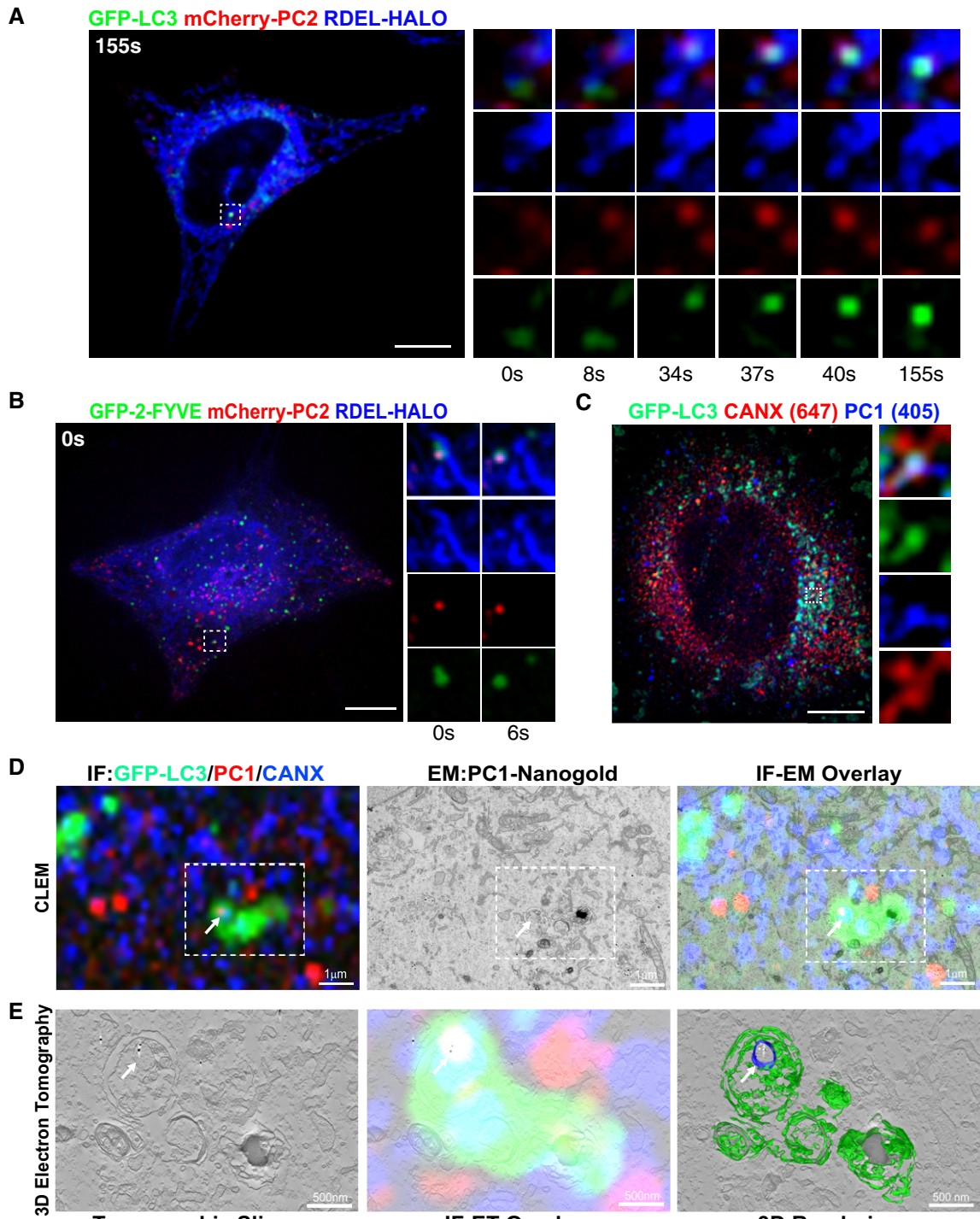

**Figure 2.  Autophagy sequesters PC molecules in the ER.**

A, B   U2OS expressing (A) GFP-LC3 or (B) GFP-2-FYVE (green), mCherry-PC2 (red) and RDEL-HALO (blue) were imaged live by spinning disc microscopy. Single and merge channels time-lapse stills at higher magnification (A = x3.93; B = x3.42) from the boxed region are shown on the right. Scale bar = 10 μm.

C   Airyscan analysis of Saos2 cells expressing GFP-LC3 (green) and immunolabelled for PC1 (405, blue) and CANX (647, red). The insets show higher magnification (x5.26) and single colour channels of the boxed area. Scale bar = 10 μm.

D   Correlative light electron microscopy (CLEM) and electron tomography of Saos2 cells transfected with GFP-LC3 (green) and labelled for PC1 (568, red and nanogold particles) and CANX (647, blue). Cells were first imaged by confocal microscopy (top left panel), and then, the same region was retraced in EM (upper middle panel) and overlay is shown (upper right panel). Arrow indicates a LC3-positive vesicle containing CANX and PC1 molecules.

E   Single tomography slice (left panel, taken from boxed are in D at a magnification of x2.84), overlay with immunofluorescence (IF) (central panel) and IF 3D rendering of AV (green) and the CANX positive vesicle containing gold particles of labelled collagen (blue and white, respectively) inside an AV (right panel).

of the PC2 triple helix and accumulation within the ER of chondrocytes. The mutations cause a type II collagenopathy in humans, named spondyloepiphyseal dysplasia congenita (SEDC) (Murray *et al*, 1989). When expressed in chondrocytes, the R789C and G1152D mutants were targeted to the lysosomes at higher rates compared with WT COL2. Notably, pharmacological enhancement of autophagy with the autophagy inducing peptide Tat-BECLIN-1 (Shoji-Kawata *et al*, 2013) increased targeting of WT and of mutant PC2 molecules to lysosomes. Opposite results were observed by treating cells with the autophagy inhibitor SAR405 (Fig EV3C). Taken together, these data suggest that autophagy preferentially degrades non-native PC molecules and prevents their accumulation in the ER.

**FAM134B is required for autophagy of procollagen**

Distinct autophagy-related (ATG) proteins and receptors play an essential role in autophagosome formation and cargo recognition, respectively (Suzuki *et al*, 2017). To characterize the machinery that enables the delivery of PC molecules to lysosomes, we silenced genes belonging to different functional autophagy clusters in Saos2 cells treated with BafA1 and quantified the levels of PC1 within lysosomes. As expected, we found that the silencing of all genes tested involved in AV biogenesis significantly inhibited the delivery of PC1 to the lysosomes. Notably, among autophagy and ER-phagy receptors, we found that *FAM134B* silencing most effectively inhibited PC1 delivery to lysosomes (Fig 3A). Our siRNA data were further validated using MEFs knocked out for genes involved in AV biogenesis, namely *Fip200* (*Fip200*$^{-/-}$), *Atg7* (*Atg7*$^{-/-}$) or *Atg16l*$^{-/-}$ as well as in MEFs lacking *Fam134b* expression (CRISPR *Fam134b*) (Figs 3B and EV4A). The effect of *Fam134b* knockout was specific, since MEFs lacking *Sec62* expression (CRISPR *Sec62*), a different ER-phagy receptor (Fumagalli *et al*, 2016), showed a normal rate of PC1 delivery to the lysosomes (Fig 3B, bottom panels). Western blot and immunofluorescence analyses confirmed the accumulation of intracellular PC1 in CRISPR *Fam134b* MEFs compared to control cells (Fig 4A and B). Notably, there was not a generalized accumulation of other ER proteins (VAPA, Sec23a and the soluble ER chaperone protein disulphide isomerase [PDI]) (Fig EV4B). The impaired delivery of PC1 to lysosomes in CRISPR *Fam134b* MEFs was rescued by reintroducing WT human FAM134B, but not a FAM134B protein lacking the (LIR) motif (FAM134Blir), in which interaction with LC3 is abolished (Khaminets *et al*, 2015) (Fig 4C). Taken together, these data strongly suggest a primary role of FAM134B in mediating the delivery of ER-resident PC molecules to lysosomes.

**Calnexin is required for autophagy of procollagen**

FAM134B is not predicted to have an ER luminal domain, so a direct interaction with PC molecules in the ER is unlikely. We also hypothesized that the PC molecules destined for degradation need to be selectively recognized by ER quality control machinery in order to be subjected to FAM134B-mediated ER-phagy. Thus, we investigated the involvement of ER chaperones in autophagy of PC. Taking advantage of a published list of putative PC1 and FAM134B ER interactors (DiChiara *et al*, 2016; Grumati *et al*, 2017), we silenced different ER genes by RNAi. The silencing of the transmembrane chaperone CANX most effectively inhibited the delivery of PC1 to lysosomes in Saos2 cells treated with BafA1 (Fig 5A).

Similar to what we observed in CRISPR *Fam134b* MEFs, *Canx*$^{-/-}$ MEFs showed an accumulation of intracellular PC1 but not of other ER-resident proteins (VAPA, Sec23a and PDI; Figs 5B and EV4C). When WT MEFs were treated with BafA1, the intracellular PC1 levels increased as consequence of defective lysosomal degradation (Fig 5B). Conversely, in *Canx*$^{-/-}$ MEFs the accumulation of PC1 was evident even in the absence of BafA1 treatment (Fig 5B). MEFs lacking *Canx* or *Crt* (*Calreticulin*) expression had an impaired PC1 delivery to lysosomes (Fig 5C). Similarly, MEFs lacking ERp57, a protein disulphide isomerase that cooperates with CANX and CRT to ensure a proper folding of proteins (Oliver *et al*, 1999), also showed a defective PC1 delivery to lysosomes (Fig 5C). The binding of CANX and CRT to target substrates occurs through the recognition of monoglucosylated oligosaccharide residues generated either by ER glucosidases I and II or by UDP-glucose: glycoprotein glucosyltransferase (UGT1) proteins (Hebert *et al*, 1995; Keller *et al*, 1998; Soldà *et al*, 2007). Pharmacological inhibition of glucosidase activities with castanospermine (CST) or deletion of Ugt1 in MEFs also inhibited PC1 delivery to lysosomes (Fig 5C). Taken together, these data indicate that all the components of the CANX/CRT cycle are required to operate the PC folding quality control and to select the misfolded PC destined to autophagy.

**Procollagens are the main substrates that accumulate in**
***Fam134b*$^{-/-}$ and *Canx*$^{-/-}$ cells**

We performed quantitative proteome analysis using mass spectrometry (MS) label-free protein quantification approach in *Canx*$^{-/-}$ and *Fam134b*$^{-/-}$ MEFs versus wild-type MEFs. *Canx*$^{-/-}$ and *Fam134b*$^{-/-}$ samples were prepared and run in parallel in order to minimize the variability due to the MS calibration and sample preparation. We identified 95 upregulated and 142 downregulated proteins in *Fam134b*$^{-/-}$ MEFs. Specifically, both Col1a1 and Col1a2 peptide chains were among the most significantly increased (−Log Student's *t*-test *P*-value: 8.55 and 8.2, respectively, for Col1a1 and Col1a2; Fig 6A and Dataset EV1). Gene ontology analysis confirmed the accumulation of collagens in MEFs lacking Fam134b (Fig EV4D). In *Canx*$^{-/-}$ cells, we identified 384 upregulated and 278 downregulated proteins. Col1a1 and Col1a2 peptides were identified as significantly increased also in *Canx*$^{-/-}$ MEFs (−log Student's *t*-test *P*-value: 3.06 and 2.37, respectively, for Col1a1 and Col1a2; Fig 6B, Dataset EV1). Interestingly, only 17 identified peptides were commonly upregulated in both *Fam134b*$^{-/-}$ and *Canx*$^{-/-}$ MEFs. Among these, collagens (Col1a1, Col1a2, Col6a1, Col6a2, Col5a1) and collagen interacting proteins (procollagen C-endopeptidase enhancer 1, SPARC/osteonectin) were the most represented categories (Fig 6C). These data clearly show that FAM134B and CANX are important regulators of PC proteostasis and that they might cooperate for the selective removal of misfolded procollagens in the ER.

**A CANX-FAM134B ER-phagy complex acts as PC autophagy receptor**

Mass spectrometry analysis identified CANX as a putative FAM134B interactor (Grumati *et al*, 2017). We confirmed this interaction by co-immunoprecipitation experiments (Fig 7A and B). CANX has a N-terminal ER luminal domain, a single transmembrane helix and a short acidic cytoplasmic tail. FAM134B instead is composed of a

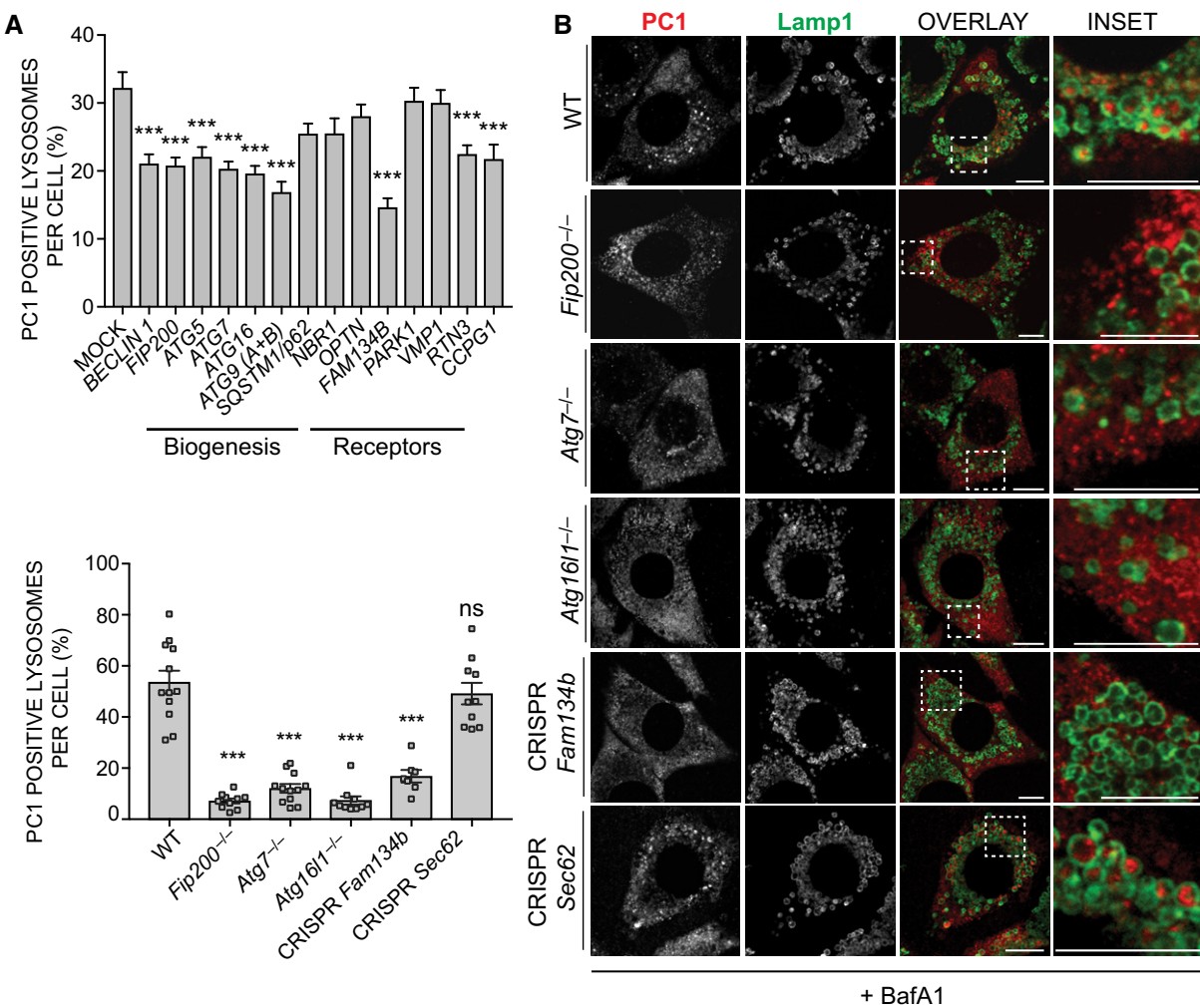

**Figure 3.  FAM134B is required for autophagy recognition of PC1.**

A  Bar graph shows quantification of lysosomes (LAMP1$^+$) containing PC1 expressed as % of total number of lysosomes (mean ± SEM) in Saos2 cells mock transfected or transfected with siRNA against the indicated genes and treated with 100 nM BafA1 for 9 h. $n$ = 20 cells per condition; three independent experiments. One-way ANOVA with Dunnett's multiple comparisons test performed, ***$P$ < 0.0001.

B  MEF cell lines lacking the expression of indicated genes were treated for 12 h with 50 nM BafA1, fixed and immunolabelled for PC1 (568, red) and LAMP1 (488, green). Scale bar = 10 μm. Insets show magnification of the boxed area. Bar graph (on the left) shows quantification of LAMP1 vesicles positive for PC1, expressed as % of total lysosomes (mean ± SEM), $n$ = 12, 10, 12, 10, 7, 10 cells per genotype, respectively; three independent experiments. One-way ANOVA with Dunnett's multiple comparisons test performed and $P$-value adjusted for multiple comparisons. ns ≥ 0.05, ***$P$ < 0.0001.

N-terminal cytosolic domain, a reticulon homology domain (containing alpha helices and a cytosolic loop) and a C-terminal cytosolic domain (Fig 7A). Thus, CANX and FAM134B could potentially interact either in the cytosol or in the ER membrane. We found that the interaction between CANX and FAM134B is lost when co-immunoprecipitation experiments were performed using a mutant version of FAM134B that lacked the intramembrane part of the reticulon homology domain, suggesting that FAM134B interacts with CANX in the ER membrane (Fig 7A and B). Notably, the FAM134Blir mutant still interacts with CANX in co-immunoprecipitation experiments (Fig 7A and B).

FAM134B-CANX interaction was not modulated by PCs, since it occurs also in HeLa (Kyoto) cells that do not express significant amounts of collagens (Hein *et al*, 2015; Fig EV5A). Functionally,

CANX is not required for FAM134B-mediated ER-phagy, as FAM134B is recruited to LC3-positive vesicles with the same efficiency in both *Canx*$^{-/-}$ and WT MEFs (Fig EV5B and C). We postulated that FAM134B interacts with misfolded PC molecules via CANX. To test this hypothesis, we generated a human osteosarcoma cell line (U2OS) expressing PC2 molecules tagged with HALO at the N terminus. HALO-PC2 was normally secreted and, similarly to endogenous PC2, accumulated in lysosomes upon BafA1 treatment (Fig EV5D and E) indicating that the presence of the HaloTag did not alter the intracellular processing of PC2. HA-resin-mediated pull-down experiments using HA-tagged FAM134B or FAM134Blir showed that both HALO-PC2 and CANX interact with FAM134B, irrespective of whether it contained the LIR domain or not (Fig 7C). Conversely co-precipitation of LC3II was dependent on a functional

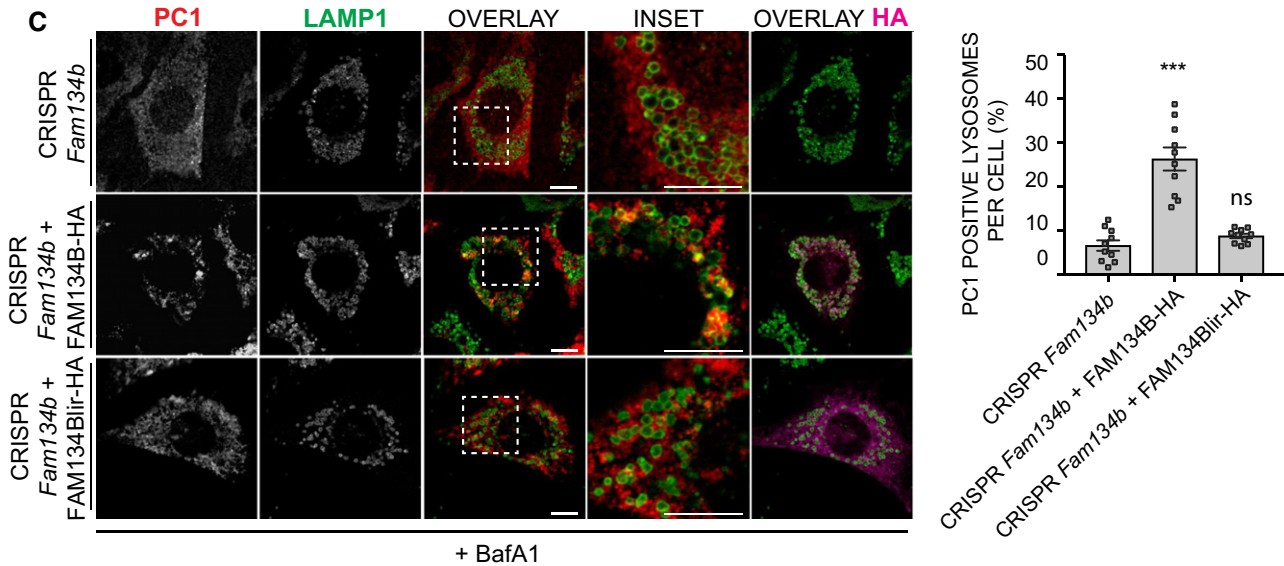

**Figure 4. PC1 accumulates intracellularly in cells lacking Fam134b.**

A   WT and CRISPR-Cas9 *Fam134b* knockout MEFs were treated as indicated, lysed and analysed by Western blot with the indicated antibodies. Western blots are representative of 4 independent experiments.

B   WT and CRISPR-Cas9 *Fam134b* MEFs were immunolabelled for PC1 (568, red), nuclei stained with Hoechst (blue) and analysed by scanning confocal microscopy. Scale bar = 10 μm.

C   CRISPR *Fam134b* MEF mock, wild-type FAM134B-HA or FAM134Blir-HA transfected were immunolabelled for PC1 (568, red), Lamp1 (488, green) and HA (647, violet) and analysed by scanning confocal microscopy. Scale bar = 10 μm. Inset panels show magnification of the boxed area. Bar graph shows quantification of Lamp1 vesicles positive for PC1, expressed as % of total lysosomes (mean ± SEM), quantification of *n* = 10 cells per treatment; three independent experiments. One-way ANOVA with Dunnett's multiple comparisons test was performed. ns ≥ 0.05, ***$P < 0.0001$.

Source data are available online for this figure.

LIR motif in FAM134B (Fig 7C), consistent with previous results (Khaminets *et al*, 2015). CST treatment diminished the level of HALO-PC2 co-precipitated by FAM134B-HA (Fig 7C) without perturbing the co-precipitation of CANX and LC3II. Taken together, these data suggest a model by which the interaction of PC with FAM134B is mediated by CANX and that the selective degradation of PC mediated by FAM134B is dependent on PC binding to CANX.

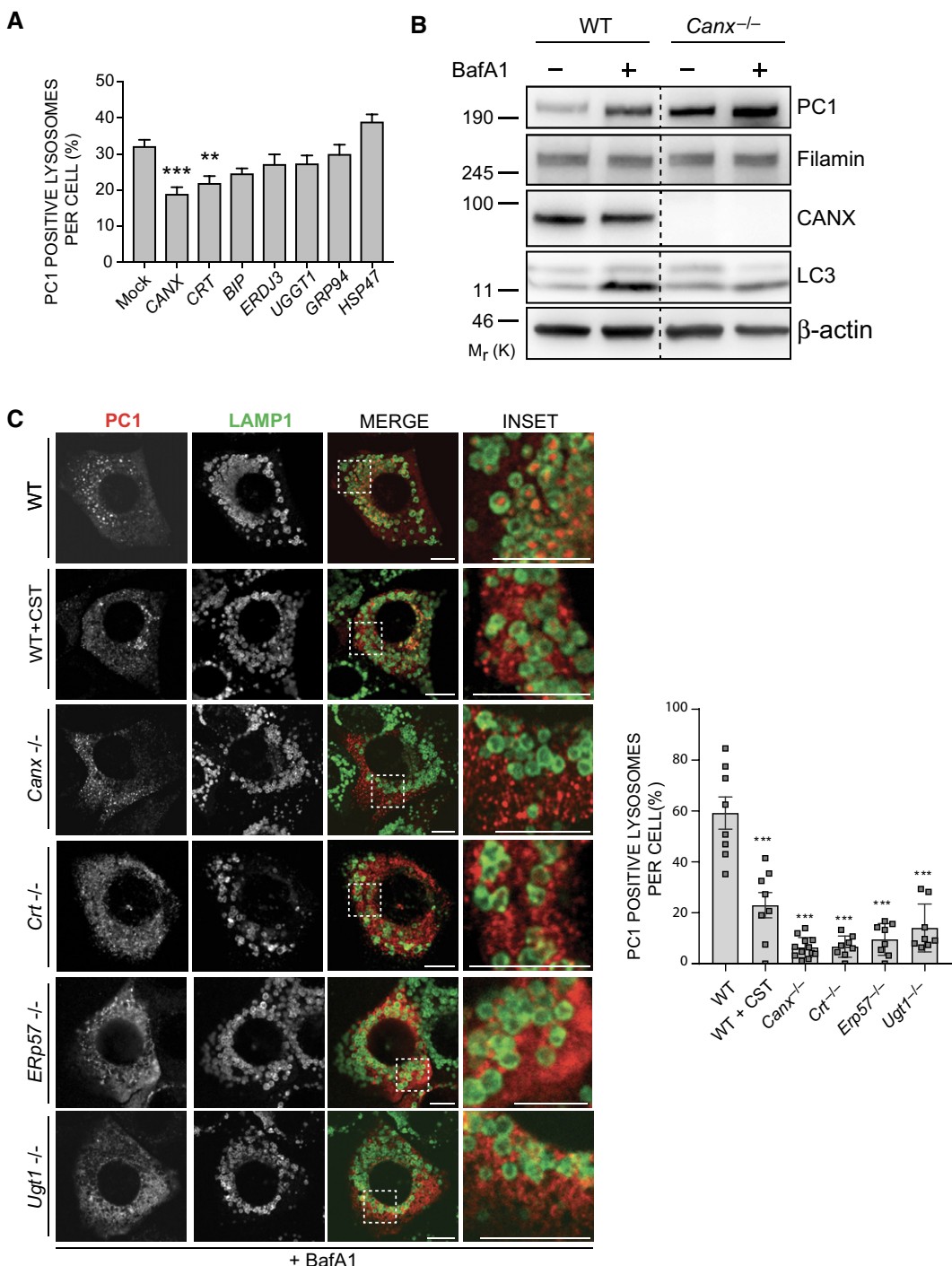

**Figure 5.  CANX is required for autophagic targeting of PC1.**

A    Bar graph shows quantification of lysosomes (LAMP1$^{+}$) containing PC1 expressed as % of lysosomes (mean ± SEM) in Saos2 cells mock transfected or transfected with siRNA against the indicated genes, treated with 100 nM BafA1 for 9 h. $n$ = 18 cells/treatment; three independent experiments. One-way ANOVA with Dunnett's multiple comparisons test performed, **$P$ < 0.005, ***$P$ < 0.0001.

B    WT and $Canx^{-/-}$ MEFs were untreated or treated with BafA1 (10 μM) for 6 h, lysed and analysed by Western blot with indicated antibodies. Filamin and β-actin were used as loading control. Dashed line indicates that unnecessary lanes were removed. Western blot is representative of three independent experiments.

C    MEF cell lines lacking the indicated genes were treated for 12 h with 50 nM BafA1 fixed and immunolabelled for PC1 (568, red) and LAMP1 (488, green). CST was added where indicated. Scale bar = 10 μm. Inset panels show magnification of the boxed area. Bar graph on the right shows quantification of LAMP1 vesicles positive for PC1, expressed as % of total lysosomes (mean ± SEM), $n$ = 8, 8, 12, 8, 8 cells, respectively; three independent experiments. One-way ANOVA with Dunnett's multiple comparisons test performed and $P$-value adjusted for multiple comparisons. ***$P$ < 0.0001.

Source data are available online for this figure.

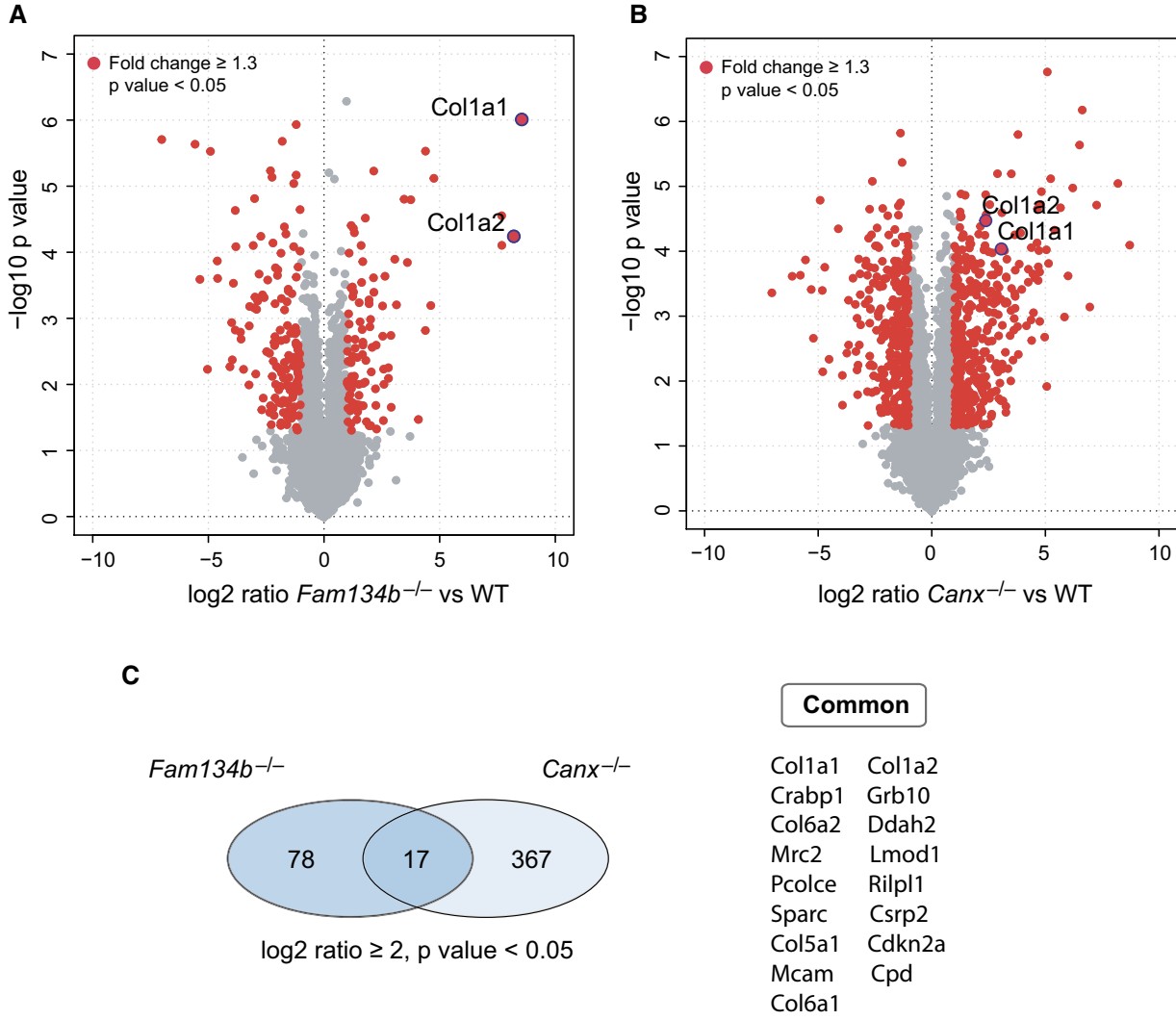

**Figure 6. PCs are the main substrates of CANX-FAM134B complex.**

A, B  Volcano plot comparing protein fold changes between WT versus *Fam134b*$^{-/-}$ (A) and WT versus *Canx*$^{-/-}$ MEFs (B). Significantly regulated proteins are labelled in red (log$_2$ fold change > 1, −log$_{10}$ P > 1.3). Red dots with blue ring indicate collagen 1 peptides. Graphs represent statistics from three separate experiments for each genotype.

C  Left: Venn diagrams represent the number of identified peptides significantly enriched in *Fam134b*$^{-/-}$ and *Canx*$^{-/-}$ MEFs. Right: List of peptides upregulated in both *Fam134b*$^{-/-}$ and *Canx*$^{-/-}$ MEFs.

# Discussion

In this work, we have investigated the mechanism by which autophagy selectively recognizes PC molecules destined for degradation in the ER. We have shown that the ER transmembrane chaperone CANX, by interacting with FAM134B and LC3, forms a novel ER-phagy complex with specific protein targeting capabilities. This complex is responsible for a specific ER clearance mechanism of PCs and links a non-native large protein within the ER lumen to the cytosolic autophagy machinery.

Firstly, we have found that silencing genes belonging to functionally different complexes involved in autophagosome biogenesis inhibited lysosomal delivery of PCs, indicating that autophagy is mediating the delivery of PC to lysosomes.

Secondly, we have identified the ER-resident autophagy receptor FAM134B as a key mediator of PC delivery to the lysosomes. FAM134B was recently identified as an ER-phagy receptor that mediates turnover of portions of the ER via autophagy (Khaminets *et al*, 2015). Our data suggest that FAM134B-dependent ER-phagy also functions as an ER quality control pathway for PCs. Our quantitative proteomic analysis in *Fam134b*$^{-/-}$ MEFs suggests that PCs are the main clients of FAM134B-mediated ER-phagy. The identification of multiple ER-phagy receptors also suggests that different cargoes might be subjected to different types of ER-phagy. Notably, a recent study showed that the disruption of CCPG1-mediated ER-phagy leads to the accumulation of ER insoluble proteins in acinar cells (Smith *et al*, 2018).

Thirdly, we have demonstrated that the chaperone CANX is a key player in PC disposal via ER-phagy. CANX is a molecular

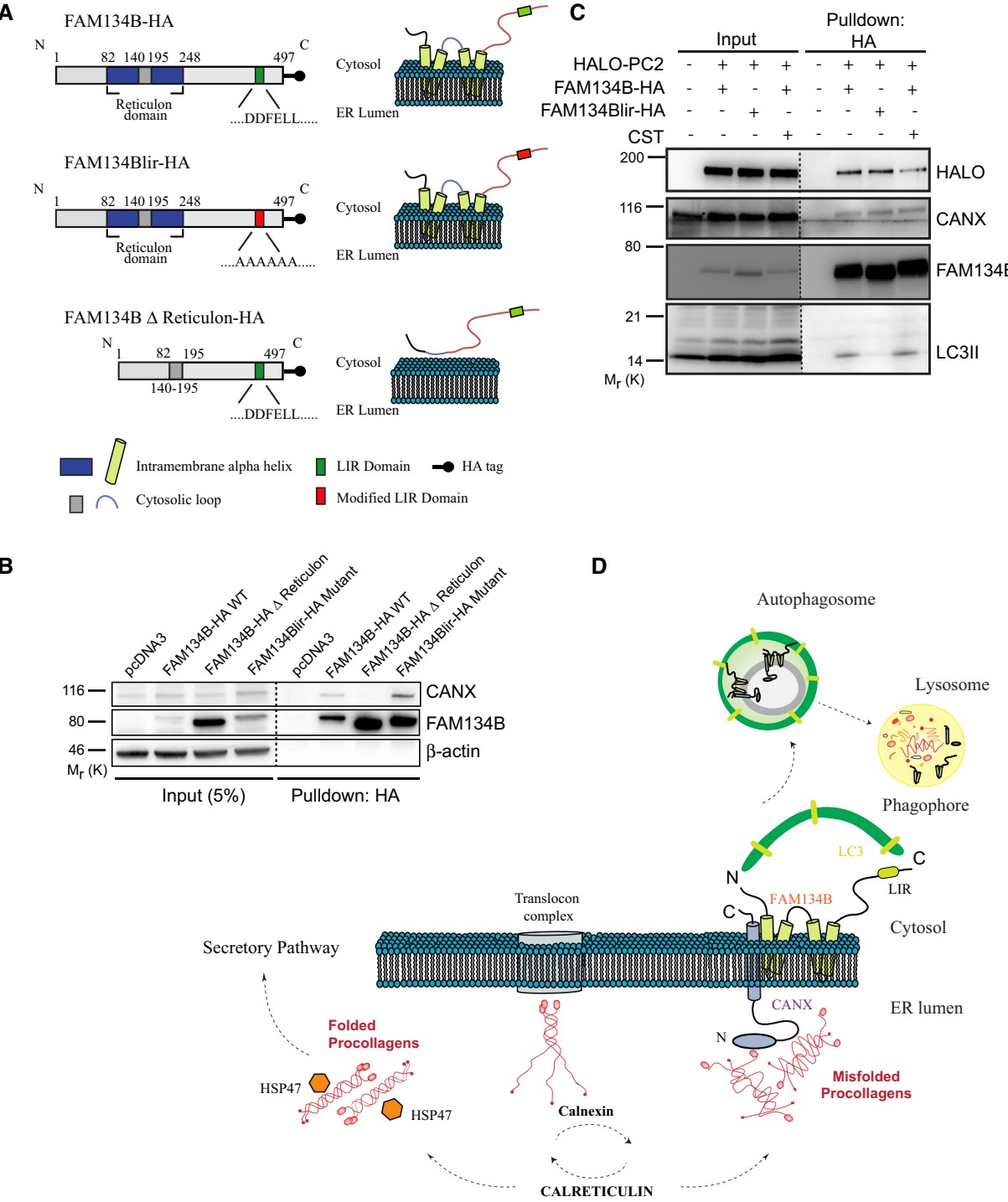

**Figure 7. CANX and FAM134B interact and deliver PC to autophagosomes.**

A   Schematic representation of FAM134B WT, lir mutant and delta reticulon constructs.

B   U2OS cells were transfected with empty vector control (pcDNA3), FAM134B-HA WT or mutant constructs as indicated.

C   U2OS cells were transfected with HALO-PC2, FAM134B-HA or FAM134Blir-HA constructs, treated with 100 nM BafA1 for 6 h and with CST where indicated. (B, C) Complexes were immune-isolated with HA-magnetic beads, separated by Western blot and visualized with antibodies against HALO, CANX, FAM134B, LC3 and β-actin. 5% of the input is shown. Western blots are representative of three independent experiments. Dashed line indicates that unnecessary lanes were removed.

D   Proposed model of collagen recognition by autophagy. After synthesis, PC chains are subjected to quality control operated by the CANX/CRT chaperone system. Properly folded PC associates with HSP47 and is then secreted, whereas the misfolded fraction is sequestered by the CANX-FAM143B complex and delivered to lysosomes through ER-phagy.

Source data are available online for this figure.

chaperone that assists the folding of monoglucosylated glycoprotein in the ER. CANX forms transient but relatively stable complexes with unfolded ER proteins until they either become folded or are degraded (Williams, 2006). The genetic or pharmacological inhibition of ER enzymes that mediate the binding of substrates to CANX impairs the delivery of PC to lysosomes, suggesting that the N-glycans-mediated recognition of PC by CANX (and CRT) represents a prerequisite for PC targeting to autophagosomes. Consistently co-immunoprecipitation experiments demonstrate that PC2 binding to FAM134B complex depends on CANX substrate affinity, since it can be reduced by CST treatment.

It is currently unknown whether additional ER partners also aid CANX. For example, ERp29, a CANX binding protein, has recently been shown to mediate the retention of immature PC1 in the ER (DiChiara *et al*, 2016).

Finally, we have also provided biochemical evidence indicating that FAM134B interacts with CANX. This interaction seems to occur within the ER membrane since it is mediated by the transmembrane regions of the reticulon homology domain of FAM134B. The reticulon homology domain generates membrane curvature by increasing the area of the cytoplasmic leaflet (Zurek *et al*, 2011). The observation that CANX-FAM134B binding is rather stable and not modulated by PCs suggests that the binding of PC to CANX might induce a conformational change of the FAM134B reticulon homology domain that increases ER membrane curvature, favouring vesicle formation. Indeed, CLEM analysis confirmed the presence of both PC molecules and CANX within a small vesicle contained within a large autophagosome, supporting the model by which portions of the ER containing both CANX and PC1 are sequestered into AVs (Fig 7D).

Mass spectrometry analyses clearly show that the CANX-FAM134B interplay is devoted to the degradation of different types of collagens, suggesting that cells may have evolved a specific mechanism to cope with the difficulties associated with the production and secretion of procollagens in the ER. This is not surprising if we consider that collagens are the most abundant proteins of our body (about 25% of our dry weight), and that its production represents a major task for cells.

We have recently reported that CANX delivers proteasome-resistant polymers of alpha 1-antitrypsin Z (ATZ) to ER subdomains *en route* for FAM134B-mediated vesicular transport to the lysosomes for degradation (Fregno *et al*, 2018). ATZ clearance, however, shows substantial differences compared to the quality control autophagy of endogenous PC that we studied in collagen-producing cells. In particular, delivery of PC molecules to lysosomes fully relies on components of the autophagosome biogenesis machinery. Conversely, many of them (e.g. ULK1/2, ATG9 and ATG13) are dispensable for ATZ clearance, suggesting that the CANX-FAM134B complex can mediate ER cargo clearance though different vesicular pathways.

Both in quality control autophagy of PC1 and ATZ clearance, the lectin chaperone CANX delivers the misfolded cargo in ER subdomains to be cleared from cells on stable interactions with FAM134B. However, other components of the CANX chaperone system (i.e. CRT, UGT1 and ERp57) cycle are required in quality control autophagy of PC1, but are dispensable for ATZ clearance.

Cumulating evidence delineates a scenario where multiple catabolic pathways ensure efficient removal of misfolded proteins from the ER lumen, which is crucial to maintain the function of this biosynthetic organelle. ER-associated-degradation (ERAD) collectively defines the many client-specific pathways engaged by misfolded proteins generated in the ER for delivery at, and dislocation across the ER membrane preceding clearance by cytosolic proteasomes (Preston & Brodsky, 2017). An increasing number of faulty gene products are shown to be excluded from the ERAD pathways (Noda & Farquhar, 1992; Fujita *et al*, 2007; Ishida *et al*, 2009; Hidvegi *et al*, 2010; Houck *et al*, 2014; Fregno *et al*, 2018). The vast heterogeneity of gene products synthesized in the ER lead us to predict that, like the multiple pathways operating for ERAD, client-specific pathways also ensure delivery of proteasome-resistant misfolded proteins to specialized ER subdomains that are eventually transported to lysosomal compartments for ER-to-lysosome-associated degradation (ERLAD).

Our results highlight the complexity of quality control pathways operating in mammalian cells to surveil the ER lumen and prevent accumulation of toxic by-products of protein biogenesis.

Lack of ER homeostasis and protein accumulation has been shown to be an underlying cause for various diseases, opening-up this pathway for development as a potential therapeutic target.

# Materials and Methods

### Cell culture, transfections, siRNA and plasmids

*Cell culture*

MEFs and RCS cell lines were cultured in DMEM with 10% FBS and 1% penicillin/streptomycin at 37°C in 5% $CO_2$. Saos2 and U2OS cells were purchased from ATCC and cultured in McCoy's medium with 15% (Saos2) or 10% (U2OS) FBS and 1% penicillin/streptomycin at 37°C in 5% $CO_2$. For collagen experiments, medium was supplemented with 50 μg/ml ascorbic acid. Wild-type, $Atg5^{-/-}$ and $Atg7^{-/-}$ MEFs were gifts from M. Komatsu and N. Mizushima. $Atg16^{-/-}$ MEFs were from T. Saitoh. $Fip200^{-/-}$ MEFs were from J.L. Guan. The generation of the *Sec62* CRISPR-Cas9 knockout MEF cell line was previously described (Fumagalli *et al*, 2016). $Canx^{-/-}$ MEF cell lines were previously described (Kraus *et al*, 2010). *Fam134b* CRISPR-Cas9 MEF cell line was described in Fregno *et al* (2018). $Fam134b^{-/-}$ MEF cell line was described in Khaminets *et al* (2015). $Crt^{-/-}$, $Erp57^{-/-}$ and $Ugt1^{-/-}$ were previously described (Molinari *et al*, 2004; Soldà *et al*, 2006, 2007).

Saos2 IDUA CRISPR-Cas9 cell line was generated as follows: For construction of the guideRNA-Cas9 plasmid, pSpCas9(BB)-2A-Puro plasmid (PX459) was obtained from Addgene (Plasmid #62988). To clone the guide sequence into the sgRNA scaffold, two annealed oligonucleotides (5′-CACCGCAGCTCAACCTCGCCTATG-3′, 5′-AAA CCATAGGCGAGGTTGAGCTGC-3′) were inserted into the pSpCas9 (BB)-2A-Puro plasmid using BbsI restriction site. Saos2 cells were transfected with the plasmid using Lipofectamine LTX and Plus Reagent (Invitrogen, Thermo Fisher Scientific) following a reverse transfection protocol. Two days after transfection, the medium was supplemented with 1 μg/ml puromycin. Puromycin-resistant clones were isolated, and gene KO was verified by sequencing. CRISPR-*IDUA* and WT Saos2 cells were kept in medium containing 1 mg/ml dermatan sulphate (Sigma-Aldrich) for 48 h before any experiment was performed.

*Transfection*

Cells were reverse-transfected using Lipofectamine LTX and PLUS reagent (Invitrogen) according to manufacturer's instructions. In Fig 4C, cells were transfected with JetPrime transfection reagent (PolyPlus) following the manufacturer's protocol. For siRNA experiments, siGENOME SMARTpool siRNAs (Dharmacon Thermo Scientific) were transfected to a final concentration of 100 nM and cells harvested 72 h after transfection.

*Plasmids*

GFP–LC3 was from Dr. Yoshimori. GFP-2-FYVE was a gift from Dr. S. Tooze. FAM134Blir-HA and FAM134BΔReticulon-HA expression plasmids were described in Khaminets *et al* (2015). HALO-PC2 plasmid was generated as follows: pLT007, a vector for CMV promoter-driven expression of N-terminally HaloTagged Col2a1, was created by replacing the mCherry tag with the HaloTag in the mCherry-C2-COL2A1 plasmid (Venditti *et al*, 2012). Standard techniques were used for construction, transformation and purification of plasmids. FAM134B-GFP was previously described (Khaminets *et al*, 2015). Site-directed mutagenesis plasmids: R789C and G1152D mutations were created using the Agilent QuikChange XL Site-Directed mutagenesis kit using the mCherry-PC2 backbone.

Primer sequences were designed with PrimerX online software and were as follows: R789C forward: 5′ CGGTCTGCCTGGG CAATGTGGTGAGAGAGGATTC 3′ and reverse: 5′ GAATCCTCTCT CACCACATTGCCCAGGCAGACCG 3′; G1152D forward: 5′GGTC CTTCTGGAGACCAAGATGCTTCTGGTCCTGCTGG 3′ and reverse: 5′ CCAGCAGGACCAGAAGCATCTTGG TCTCCAGAAGGACC 3′.

**Immunofluorescence**

Cells were seeded on coverslips at least 24 h before treatment and fixed for 10 min in 4% PFA, or for the detection of endogenous LC3, fixed for 10 min in ice-cold methanol. Cells were blocked and permeabilized for 30 min in blocking buffer (0.05% (w/v) saponin, 0.5% (w/v) BSA, 50 mM $NH_4Cl$ and 0.02% $NaN_3$ in PBS, pH 7.4). For LAMP1 immunolabelling, 15 mM glycine was added to blocking buffer. Cells were incubated for 1 h with the following primary antibodies: collagen I (SP1.D8, Hybridoma Bank), collagen II (II-II6B3; Hybridoma Bank), LAMP1 (Abcam, ab24170) or LAMP1 (Hybridoma Bank, 1D4B was deposited to the DSHB by August, J.T.), CANX (Enzo Life Sciences ADI-SPA-860-D), LC3 (NB100-2220; Novus Biologicals), HSP47 (Abcam, ab77609); HA (Sigma, H6908), washed 3 times in PBS; incubated for 45 min with secondary antibody (Alexa Fluor–labelled goat anti-rat A11077, goat anti-guinea pig A11073, goat anti-rabbit A11011/A11008, and goat anti-mouse A11001, A11004; Life Technologies, Thermo Fisher Scientific); washed three times in PBS; incubated for 20 min with 1 μg/ml Hoechst 33342, and finally mounted in Mowiol (Sigma-Aldrich) or Vectashield (Vector Laboratories) supplemented with 4′,6-diamidino-2-phenylindole (DAPI).

**Medaka stocks**

Samples of the Cab strain of wild-type medaka fish were kept and staged as described previously (Iwamatsu, 2004; Carrella *et al*, 2015). All studies on fish were conducted in strict accordance with the institutional guidelines for animal research and approved by the

Italian Ministry of Health; Department of Public Health, Animal Health, Nutrition and Food Safety in accordance to the law on animal experimentation (article 7; D.L. 116/92). Furthermore, all animal treatments were reviewed and approved in advance by the Ethics Committee at the TIGEM Institute [Pozzuoli (NA), Italy].

**Immunofluorescence analysis in Medaka fish embryos**

The animals were subjected to anaesthesia before fixation at stage 40 by 2 h of incubation in methanol 100% at room temperature (RT). Samples were rinsed three times with PTw 1× (1× PBS, 0.1% Tween, pH 7.3) and then incubated overnight in 15% sucrose/PTW1X at 4°C, and then again incubated overnight in 30% sucrose/PTW1X at 4°C. Cryosections of the larvae were processed for immunostaining as follows: rehydrated in 1× PBS for 30 min, washed in PBS-0.1% Triton X-100 and treated with antigen retrieval solution [proteinase K 20 mg/ml (Sigma-Aldrich, Germany) dissolved in 10 mM Tris pH 8.0, 1 mM EDTA (TE)] for 15 min at 37°C. Cryosections were then permeabilized with 0.5% Triton X-100 in 1× PBS for 20 min at RT, rinsed in PBS 0.1% Triton X-100 and moved to blocking solution [2% BSA, 2% serum, 2% DMSO in PBS-0.1% Triton X-100] for 30 min at RT. Cryosections were incubated with rabbit anti-collagen type II (Rockland, 1:400) and mouse anti-LC3B (Nanotools, 1:100) antibodies overnight at 4°C, then washed with PBS-0.1% Triton X-100 and incubated with secondary antibodies, Alexa-594 anti-rabbit IgG (1:500), Alexa-488 anti-mouse IgG (1:500; Thermo Fisher) for 1 h at RT. Nuclei were stained with DAPI (1:500).

**Chemicals and cell treatments**

L-Ascorbic acid (Sigma-Aldrich) was made fresh and used at a final concentration of 50 μg/ml from the beginning of each experimental procedure. Bafilomycin A1 (BafA1; Sigma-Aldrich) was used at a final concentration of 100 nM, and compared to DMSO (Sigma-Aldrich) as vehicle for 6 h (RCS) or 9 h (Saos2/U2OS). MEFs were treated with 50 nM bafilomycin for 12 h or 100 nM for 6 h. Castanospermine (CST; Sigma-Aldrich) was used at a final concentration of 1 mM. CST was added 2 h before BafA1, and ascorbic acid treatment. SAR405 (Selleckchem) was used at a concentration of 10 μM for 2 h preceding and throughout BafA1 treatment. Tat-BECLIN-1 (D17, Millipore) was used at 5 μM in acidified media for 4 h then replaced with fresh media for 2 h before harvesting cells. HaloTag, far red (ex. 650 nm, em. 668 nm) SiR HaloTag ligand (Promega), available through custom order, incubated in media at 2 mM for 3 h. 0.5 μM TMR (Promega) was added to the media 2 h pre-fixation for lysosome visualization, or for pulse chase, 20 min at 1 μM, followed by p5030 (Promega). Rutin (Acros Organics) was used at 10 μg/ml for the duration of live cell imaging.

**Confocal microscopy**

Scanning laser confocal experiments were acquired using a Zeiss LSM 800 or Leica TCS SP5 confocal microscope equipped with a 63× 1.4 numerical aperture oil objective. Airyscan microscopy was performed using a Zeiss LSM 880 confocal microscope, equipped with Plan-Apochromat 63×/1.4 numerical aperture oil objective and pixel size of 8.7 nm. Images were subjected to post-acquisition Airyscan processing. Image acquisition and processing were

performed with Zen Blue software and co-localization analysis and image presentation was performed using ImageJ FIJI software or Photoshop (Adobe).

## Live cell imaging

U2OS cells were transiently transfected with mCherry-PC2 and RDEL-HALO plus GFP-LC3 or GFP-2-FYVE. Cells were incubated on a Tokai Hit stage top incubator heated stage in 5% $CO_2$ at 40°C in the presence of far red HALO ligand for 3 h. Immediately prior to imaging, medium was supplemented with ascorbic acid and rutin (routinely used to decrease photobleaching). Imaging was initiated at temperature switch to 32°C. Frames were acquired at 1-s intervals. Imaging was performed on a Nikon Inverted Spinning Disk confocal with sCMOS Prime95B camera (Photometrics) with pixel size of 11 μm, using a 100× CFI Plan Apo oil objective with 1.4 NA. Image acquisition was performed with Metamorph 7.7.6 software (Molecular Devices, France) and processing in ImageJ FIJI software.

## Correlative light electron microscopy (CLEM) and Tomography

Saos2 cells were grown on gridded MatTek glass-bottomed dishes (MatTek Corporation) transfected with GFP-LC3 and fixed with 0.05% glutaraldehyde in 4% paraformaldehyde (PFA) and 0.1 M HEPES buffer for 10 min, washed once in 4% PFA, then incubated in fresh 4% paraformaldehyde in 0.1 M HEPES buffer for 30 min. Subsequently, cells were incubated for 30 min in blocking buffer and immunolabelled for collagen I (SP1.D8 Hybridoma Bank) and CANX (ADI-SPA-860-D Enzo Life Sciences), visualized with Alexa-Fluor546 fluoro-nanogold Fab' conjugate (Nanoprobes) and Alexa-Fluor647 Rabbit Ab, respectively. Nanogold was enlarged using gold enhancement kit (Nanoprobes) according to manufacturer's instructions. Samples were then post-fixed with 1.5% potassium ferricyanide, 1% $OsO_4$ in 0.1 M cacodylate buffer for 1 h on ice and en bloc stained overnight with 1% uranyl acetate. Samples were dehydrated in ethanol and embedded in epoxy resin (SIGMA). After baking for 48 h at 60°C, the resin was released from the glass coverslip by temperature shock in liquid nitrogen. Serial sections (70–90 nm) were collected on carbon-coated formvar slot grids and imaged with a Zeiss LEO 512 electron microscope. Images were acquired with a 2k × 2k bottom-mounted slow-scan Proscan camera controlled by EsiVisionPro 3.2 software.

For electron tomography, tilted series were acquired with a 200 kV Tecnai G2 20 electron microscope (FEI, Eindhoven) at a magnification of 11.5 k, resulting in pixel size of 1.95 nm. Single, tilted image series (± 60° according to a Saxton scheme with the initial tilt step of 2°) were acquired using Xplorer3D (FEI) with an Eagle 2,048 × 2,048 CCD camera (FEI). Tilted series alignment and tomographic reconstructions were done with the IMOD software package. Image segmentation was done by MIB software (BW thresholding) and visualized using IMOD.

## Transmission electron microscopy

Cells were fixed in 1% glutaraldehyde in 0.2 M HEPES buffer and then post-fixed in uranyl acetate and in $OsO_4$. After dehydration through a graded series of ethanol, samples were cleared in propylene oxide, embedded in epoxy resin (Epon 812) and polymerized at 60°C for 72 h. From each sample, thin sections were cut with a Leica EM UC6 ultramicrotome and images were acquired using a FEI Tecnai −12 (FEI) electron microscope equipped with Veletta CCD camera for digital image acquisition.

## Immunoprecipitation experiments

HA-tag precipitation: U2OS cells were transiently transfected with plasmids encoding HALO-PC2 and FAM134B-HA. On the day of experiment, plates were treated with 1 mM CST where indicated for 2 h, then all plates treated with 100 nM BafA1 and 50 μg/ml ascorbic acid for 4 h. Cells were detached with trypsin–EDTA and centrifuged. The cell pellets were washed three times with ice-cold PBS and then resuspended in 1 ml MCLB lysis buffer (1% NP-40, 150 mM NaCl, 50 mM Tris/HCl pH 8). The cell suspension was lysed by passing it through a 24-G needle for 10-15 times. The lysates were incubated on ice for 20 min with gentle swirling and centrifuged at 18,000 *g* to pellet nuclei and cell debris. The supernatants were collected and subjected to protein quantification using BCA protein assay kit (Pierce Chemical). 1 mg of each lysate was then precipitated using Pierce anti-HA-magnetic beads (Thermo Fisher Scientific) and rotated at 4°C overnight. The precipitated proteins were washed three times with MCLB lysis buffer (1% NP-40, 150 mM NaCl, 50 mM Tris/HCl pH 8) and two times with the same lysis buffer, detergent free. The protein complexes were resuspended in 1v/v 2× Laemmli sample buffer and analysed by SDS–PAGE in a 7–14% gradient gel.

HeLa (Kyoto) cells and U2OS cells were transiently transfected with empty vector control, FAM134B-HA WT or mutant constructs. On the day of the experiment, cells were detached with trypsin–EDTA and centrifuged. Immunoprecipitation experiments were performed in the same conditions and analysed by SDS–PAGE in a 4–15% Mini-PROTEAN® TGX™ Precast Protein gel.

## Western blot analysis

Cells were washed twice with PBS and then scraped in RIPA lysis buffer (20 mM Tris [pH 8.0], 150 mM NaCl, 0.1% SDS, 1% NP-40, 0.5% sodium deoxycholate) in the presence of PhosSTOP and EDTA-free protease inhibitor tablets (Roche). Cell lysates were incubated on ice for 30 min, and then, the soluble fraction was isolated by centrifugation at 16,000 *g* for 20 min at 4°C. The total protein concentration in cellular extracts was measured by BCA protein assay kit (Pierce Chemical). Protein extracts, separated by SDS–PAGE and transferred onto membranes, were probed with antibodies against COLLAGEN I (Abcam, ab138492 for human cells; Abcam, ab21286 for mouse cells), LAMP1 (Abcam, ab24170), LC3B (NB100-2220; Novus Biologicals), CANX (Enzo Life Sciences ADI-SPA-860-D), HALO (Promega G928A) FAM134B (Sigma, HPA012077), SQSTM1/p62 (Abnova, H00008878-M01), SEC23A (PAI-069A; Thermo Fisher Scientific), VAP-A (15275-1-AP, Proteintech), PDI (Enzo Life Sciences ADI-SPA-891-F), Filamin (Abcam, ab76289), TUBULIN (T5168, Sigma-Aldrich) and β-actin (Novus Biologicals NB600-501), probed with horseradish peroxidase (HRP)-conjugated goat anti-mouse or anti-rabbit IgG antibody (1:2,000, Vector Laboratories; 8125, 8114; Cell Signaling Technology) and visualized with the Super Signal West Dura substrate (Thermo Fisher Scientific), according to the manufacturer's protocol.

## Mass spectrometry

Wild-type, *Fam134b* and *Canx* knockout MEFs were grown in DMEM media supplemented with 10% FBS. Cells were lysate in SDS-lysis buffer (4% SDS in 0.1 M Tris/HCl pH 7.6). Protein concentration was measured using BCA Kit (Pearce), and 50 μg of cells lysate was precipitated with ice-cold acetone and resuspended in 30 μl of GnHCl buffer (6 M guanidine hydrochloride, 50 mM Tris pH 8.5, 5 mM TCEP, 20 mM chloro-iodoacetamide). For label-free quantification-based proteome analysis of whole cell lysates, proteins were in-solution digested with the endopeptidase sequencing-grade Lys-C (1:100 ratio) for 3 h at 37°C and subsequently with trypsin (1:100 ratio) overnight 37°C. Digestion was blocked with TFA 1% final concentration. Collected peptide mixtures were concentrated and desalted using the Stop and Go Extraction (STAGE) technique (Rappsilber *et al*, 2003).

Instruments for LC-MS/MS analysis consisted of a NanoLC 1200 coupled via a nano-electrospray ionization source to the quadrupole-based Q Exactive HF benchtop mass spectrometer (Thermo Scientific). Peptide separation was carried out according to their hydrophobicity on an in-house packed 20 cm column with 1.9 mm C18 beads (Dr Maisch GmbH) using a binary buffer system consisting of solution A: 0.1% formic acid (0.5% formic acid) and B: 80% acetonitrile, 0.1% formic acid (80% acetonitrile, 0.5% formic acid). 2 h gradients were used for each sample. Linear gradients from 5–38% B were applied with a following increase to 95% B at 400 nl/min and a re-equilibration to 5% B. Q Exactive HF settings: MS spectra were acquired using 3E6 as an AGC target, a maximal injection time of 20 ms and a 60,000 resolution at 200 $m/z$. The mass spectrometer operated in a data-dependent mode with subsequent acquisition of higher-energy collisional dissociation (HCD) fragmentation MS/MS spectra of the 15 most intense peaks. Resolution for MS/MS spectra was set to 30,000 at 200 $m/z$, AGC target to 1E5, max injection time to 25 ms and the isolation window to 1.6 Th.

## Statistics

Statistics were performed in GraphPad PRISM software. A two-tailed, paired and unpaired Student's *t*-test was performed when comparing the same cell population with two different treatments or cells with different genotypes, respectively. One-way ANOVA and Dunnett's post hoc test were performed when comparing more than two groups relative to a single factor (treatment). A *P*-value of 0.05 or less was considered statistically significant.

For mass spectrometry analysis, the raw files were processed using MaxQuant software (Cox *et al*, 2011). Parameters were set to default values. Statistical analysis, *t*-test and GO annotation enrichment were performed using Perseus software (Tyanova *et al*, 2016). Data are representative of three independent mass spectrometry analyses for each genotype.

**Expanded View** for this article is available online.

## Acknowledgements

We thank G. Diez Roux, A. Ballabio, C Di Malta, G. Napolitano, and L. Soria (TIGEM) for suggestions and critical reading of the manuscript; we thank C. Lanzara and L. Cinque for support and César Valades Cruz for support in live cell image processing and movie creation. The authors acknowledge Euro-BioImaging (www.eurobioimaging.eu) for providing access to imaging technologies and services via the Italian Node (ALEMBIC, Milan-Italy). The Cell and Tissue Imaging (PICT-IBiSA) of the Institute Curie (ANR10-INBS-04). This work was supported by grants to CS: Italian Telethon Foundation (TCP12008), TIGEM institutional Grant, European Research Council (ERC) starting grant (714551), and Associazione Italiana per la Ricerca sul Cancro (A.I.R.C.) (IG 2015 Id 17717); MM: Signora Alessandra, AlphaONE Foundation, Foundation for Research on Neurodegenerative Diseases, the Novartis Foundation, Swiss National Science Foundation (SNF) and Comel and Gelu Foundations. To ID: DFG-funded Collaborative Research Centre on Selective Autophagy (SFB 1177), ERC no. 742720, DFG-funded SPP 1580 program and by the LOEWE program Ubiquitin Networks (Ub-Net) funded by the State of Hesse/Germany.

## Author contributions

AF, CDL and EF performed most of the experiments. PG and MS performed mass spectrometry experiments. LS and GB performed HA-pull-down studies. AM and IF performed IF experiments in KO MEF. IC and DI performed experiments using Medaka fish. AR performed CLEM and tomography. AM, IF, MP and MI helped with the quantification of the data. EVA generated and characterized HALO-PC2 plasmids. MADM provided critical suggestions, protocols and reagents. CS designed the project. CS, MM and ID supervised the study. CS and AF wrote the manuscript.

## Conflict of interest

The authors declare that they have no conflict of interest.

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
