## [Review Process File · The EMBO Journal]

A selective ER-phagy exerts procollagen quality control via a CALNEXIN-FAM134B complex

Alison Forrester, Chiara De Leonibus, Paolo Grumati, Elisa Fasana, Marilina Piemontese, Leopoldo Staiano, Ilaria Fregno, Andrea Raimondi, Alessandro Marazza, Gemma Bruno, Maria Iavazzo, Daniela Intartaglia, Marta Seczynska, Eelco van Anken, Ivan Conte, Maria Antonietta De Matteis, Ivan Dikic, Maurizio Molinari and Carmine Settembre.

Review timeline:

Submission date:	17 th May 2018
Editorial Decision:	28 th June 2018
Revision received:	11 th September 2018
Editorial Decision:	5 th October 2018
Revision received:	10 th October 2018
Accepted:	31 st October 2018

Editor: Elisabetta Argenzio

Transaction Report:

1st Editorial Decision

28th June 2018

Thank you for submitting your manuscript on a role for autophagy in procollagen degradation through Calnexin and FAM134B to The EMBO Journal. We have now received three referee reports on your study, which are enclosed below for your information.

As you can see, while all the referees judge the findings to be overall interesting, referee #1 and #2 also raise several critical points that need to be addressed before they can support publication at The EMBO Journal. In particular, referee #1 is concerned that autophagy needs to be further characterized in your system (e.g. using additional autophagy markers, and combining imaging and western blot techniques). Also, s/he finds that the procollagen-dependent interaction between Calnexin and FAM134B requires deeper investigation, and that all biochemical and imaging experiments have to be properly quantified and analyzed for statistical significance. Referee #2 points out that the study fails to address the role of Calnexin and Calnexin/FAM134B interplay in ER-phagy and requests you to clarify the selectivity of the Calnexin-FAM134B pathway towards different cargoes and its involvement in ER stress sensing. In addition, this referee finds that the study does not provide mechanistic insight on how misfolded procollagen and Calnexin are sequestered in a given subdomain and states that the physiological relevance of the fraction of procollagen that is misfolded and degraded need to be investigated.

Addressing these issues through decisive additional data as suggested by the referees would be essential to warrant publication in The EMBO Journal. Given the overall interest of your study, I would thus like to invite you to revise the manuscript in response to the referee reports.

REFEREE REPORTS

Referee #1:

It has been recently reported that autophagy degrades unfunctional procollagen (PC), however the mechanism by which the protein is recruited into autophagosome remained largely elusive. Forrester, et al, report here that the ER-phagy receptor FAM134B together with Calnexin (CNX) are responsible for the recruitment of PC directly from the ER to autophagosome. The authors applied microscopy and biochemical approaches using mouse embryonic fibroblasts (MEF), human osteoblasts (Saos2) and osteoblasts of the mandible in Medaka fish embryos serving as the physiological system.

Overall, this is an interesting and mostly well-executed study; however there are few important issues that must be resolved to solidify the authors' model. For example, autophagy is mostly determined in this study by microscopy techniques with limited number of markers. The authors are encouraged to utilize additional markers to determine autophagy and to combine it with the well-established WB tools. More importantly, it is important to better characterize the suggested interaction namely to determine whether CNX directly interacts with FAM134B in PC dependent manner. The authors should also explain why they performed the pulldowns with PC2 while the entire study was done on PC1

Additional comments

- Add quantification and statistical information to all images.
- In pulldown experiments the input and the pulldown should be in the same gel.
- Add Western blots with autophagic markers and quantifications to all experiments shown in figures 1-3
- WB with anti-FAM antibodies is missing from the experiment shown in figure 4b, moreover some of the blots (LC3, actin and VAPA) seem overexposed.
- The experiment shown in figure 5E is missing the analysis with anti-CNX and anti-LC3 antibodies.
- How the authors explain the LC3 band shown in HALO pulldown (figure 7b). WB with anti-HALO is missing (in both 7b and c).

The data presented in figure 7c are not convincing. The changes in mobility CNX are not explained and the effect of CST is marginal.

Minor comment:

Specify the bafA treatment in the actual images may help.

Referee #2:

In this manuscript, Forrester and colleagues report a role for lysosome and autophagy in the degradation of misfolded procollagen protein mediated by the ER-phagy receptor FAM134B and the ER protein CNX (calnexin). Overall, the results sound interesting and the topic fits more or less with the general interest for the cell and molecular biology community and the readers of EMBO Journal. However, the pivotal question concerning the central role of CNX (and CNX-FAM134B interplay) in broader specialized ER-phagy is not really addressed, despite the demonstration of its importance in pro-collagen degradation. Based on this, I am not totally convinced that this story must be published in EMBO journal as it is describing (pretty well) only the pro-collagen lysosomal degradation, but has no general impact on the field of ER/autophagy interplay.

MAJOR CONCERNS:

As written in the summary section, my principal remark concerns the importance/originality of the message, regarding the fact that FAM134B-mediated ER-phagy has been nicely studied and documented previously (including by authors of the present study) and that the relation between procollagen and lysosomal degradation has been suggested already (see review by same authors, in Matrix Biol, June 2018) but without questioning the relevance of this system for other cargoes.

Indeed, as authors conclude that "the [CNX-FAM134B complex] is responsible for a specific ER-clearance mechanism that is crucial for ER and cellular homeostasis", this would be probably relevant to decipher whether or not this "FAM134B-CNX" ER pathway is indeed dedicated to a broad (or not) range of unfolded proteins turnover and to analyze this could be associated with stress situations in specific context(s) (such as bone physiology) and related to ER stress sensing. The conclusions presented in the current manuscript are most of the time well supported by experimental data, but the dynamic aspect of the described pathway is lacking: notably, the correlative light electron microscopy used in the paper (Figure 6) is not very informative since it described only one situation in time. It would have been more useful to assess the pathway dynamically, by videomicroscopy for example, to document the capture of PC-positive ER structure by pre-existing autophagosome and then its delivery to the lysosome. Moreover, there is no precise information about autophagic machinery recruitment at CNX specific zones of the ER engaged with PC handling. Are these domains positive for PI3P, DCFP1, ULK1, etc?

Finally, there is not real data on the mechanisms by which misfolded PC can be sequestered together with enriched CNX in a given subdomain of the ER: such data could highlight a dynamic process by which chaperones such as CNX could trigger spatial sorting in the ER membrane, to facilitate ER piecemeal autophagy process.

In summary (and to my opinion), this manuscript is closer of a "cell biology case report" concerning the degradation of PC by autophagolysosome (and again, most of the experiments are well addressed) than a broad cell biology message on ER-phagy and ER dynamics interplay in response to general misfolded proteins.

MINOR POINTS:

1) Regularly, authors refer to "a fraction" of PC that is misfolded, but readers lack details to understand the physiological relevance of such amount of misfolded PC: is that cell type dependant? Could it be modulated by given physiological or pathophysiological situations, including stress and autophagy inducing treatments? Could be related somehow to autophagy defects or modulating situations?

2) the experimental rationale of data presented in Figure 1 G-H is not very clear and should better explained.

3) the use of ATG9 CRISPR cell line is not clear to me: why ATG9? Usually, studies aimed at investigating implication of autophagy (or autophagic machinery) in a given process are often addressed with genes directly implicated with autophagosome biogenesis (including genes from class III PI3K such as Beclin1, ATG14 or VPS34) or LC3 lipidation regulation (such as ATG7, ATG5, ATG12 or ATG16). It is not clear why data with ATG-/- MEFs are presented independently from ATG9 data... The putative role of ATG9 in non autophagic or non-canonical autophagy related processes is not very in favour of the conclusion of authors claiming that "... autophagy is responsible for delivery of PC molecules to lysosomes for degradation". Manuscript/figures could be modified to make it clearer.

4) the discussion section could benefit of more hypotheses concerning a putative new function for ER chaperones in selective sorting of misfolded proteins for autophagy / ER-phagy.

5) authors refer to LAMP1 as a lysosomal marker, while the precise localization of LAMP1 is lysosome and late endosomes.

Referee #3:

In my view, this is an outstanding paper that uses many approaches to disclose the mechanism of autophagy quality control of procollagen (PC) via a Calnexin-FAM134B ER-Phagy complex.

The principal conclusions are that canonical autophagy mediates PC delivery from ER to lysosomes; that the ER-resident autophagy receptor FAM134B is a key regulator in the process; that Calnexin (CNX) interacts with FAM134B and they in turn interact with LC3 to form a novel ER-phagy complex which provides the clearance mechanism for delivery of PC from ER to lysosomes for turnover.

The paper substantially advances the field. It is very well written and is a pleasure to read. Experiments are well designed and data presented clearly. The combination of biochemical, cell

biological and genetic techniques is quite impressive and, in toto, provide definitive evidence to support the conclusions.

My one minor suggestion is to change the legend to Figure EV4 from a summary statement to one that describes what is shown in the Figure. Eg, Suggest it read " Figure EV4 Unlike with PC1 (Fig 5C,D), CNX deficiency fails to increase accumulation of other resident ER Proteins "

1st Revision - authors' response

11th September 2018

We have responded to the referees' comments point by point below, including listing all new experiments performed. All new text relating to new experiments is highlighted in yellow in the manuscript.

Referee #1:

It has been recently reported that autophagy degrades unfunctional procollagen (PC), however the mechanism by which the protein is recruited into autophagosome remained largely elusive. Forrester, et al, report here that the ER-phagy receptor FAM134B together with Calnexin (CNX) are responsible for the recruitment of PC directly from the ER to autophagosome. The authors applied microscopy and biochemical approaches using mouse embryonic fibroblasts (MEF), human osteoblasts (Saos2) and osteoblasts of the mandible in Medaka fish embryos serving as the physiological system.

Overall, this is an interesting and mostly well-executed study; however there are few important issues that must be resolved to solidify the authors' model. For example, autophagy is mostly determined in this study by microscopy techniques with limited number of markers. The authors are encouraged to utilize additional markers to determine autophagy and to combine it with the well-established WB tools.

We thank the referee for his/her important comments. We performed additional experiments addressing these concerns and modified the manuscript accordingly.

1. In addition to the marker LC3, we studied DFCP1, a phosphatidylinositol-3-phosphate (PI3P) binding protein by super-resolution microscopy. DFCP1 puncta represent the site where autophagosomes originate (omegasome) at the endoplasmic reticulum (Hannah C. Dooley and Sharon A. Tooze, Mol Cell 2014). We analyzed the colocalization of procollagen (PC) with DFCP1 marker at steady state, in both MEF and Saos2. We found that about 20% of DFCP1 positive structures co-localize with PC1. Data are presented in Figure EV 1A,B.
2. We have performed videomicroscopy in cells that express GFP-FYVE (PI3P binding probe), mcherryPC2, and RDEL-HALO (ER marker) and captured the biogenesis of an autophagosome containing ER and PC2 (Fig. 2B and Movie EV2).
3. To further characterize the autophagy pathway, we performed western blot analysis using PC1 and PC2, p62/SQSTM1 and LC3 markers in the experiments shown in Figure 1. Specifically, p62/SQSTM1 and LC3 accumulate along with PCs in MEF, Saos2 and RCS upon BafA1 treatment. Data are shown in Figure EV 2A. Moreover, LAMP1, LC3II and PC1 accumulate in CRISPR IDUA Saos2 by Western blot (Fig. 1J). These biochemical data clearly show that PCs are autophagy substrates.
4. We biochemically verified that MEFs lacking FIP200, ATG7 and ATG16 accumulate p62/SQSTM1 compared with control cells, and that LC3 lipidation was impaired in ATG7^{-/-} and ATG16^{-/-} MEFs. Notably, PC1 accumulates in autophagy deficient MEFs as shown by western blotting. Data are presented in Figure EV 5A. Consistent with previous results, FAM134B^{-/-} MEFs do not show any impairment of canonical autophagy (e.g. LC3II lipidation and p62/SQSTM1 accumulation) (Figure EV 5A), however, consistent with our working model, they show accumulation of PC1 (Figure 4A and Figure EV 5A).

More importantly, it is important to better characterize the suggested interaction namely to determine whether CNX directly interacts with FAM134B in PC dependent manner. The authors should also used explain why they performed the pulldowns with PC2 while the entire study was done on PC1.

The manuscript contains several experiments performed using PC2 (we have now included them in the main figures). The data clearly indicate that PC1 and PC2 behave similarly. PC2 is a trimeric assembly of a single protein chain (Col2a1), whereas PC1 is composed of two different protein chains (Col1a1 and Col2a2) assembled in a 2:1 stoichiometry. We reasoned that overexpression of one of the two Col1 genes might lead to an inefficient incorporation of the tagged protein into PC1 chains, leading to an accumulation of unassembled exogenous protein in the ER. This scenario might favor the activation of ER stress pathways, which might interfere with the interpretation of the results. For these reasons we decided to perform the overexpression studies using PC2. Consistently, our pulse and chase experiments demonstrated that HALO-PC2 was properly assembled and secreted extracellularly and accumulates in lysosomes in BafA1 treated cells (Fig. EV8 A,B).

To better characterize the interaction between CNX and FAM134B and to understand whether it is mediated by PC we performed additional experiments and found that:

- 1) FAM134B interacts with CNX through the intramembrane domain, since CNX is not immunoprecipitated by a FAM134B mutant that lacks the intramembrane parts of the Reticulon Homology Domain. Conversely the interaction is maintained in the absence of the FAM134B LIR domain (Fig. 7A,B).
- 2) The interaction between FAM134B and CNX is not dependent on PCs since CNX-FAM134B interaction can also be detected in cells (HeLa Kyoto) that do not produce significant amounts of collagens (Hein et al. Cell. 2015 Oct 22;163(3):712-23) (Fig. EV 7A).

- 3) The sequestration of FAM134B-positive ER structures into autophagosomes does not require the activity of CNX since we found no significant change in the FAM134B-LC3 co-localization extent between WT and CNX ^{-/-} MEFs (EV Fig. 7B,C)

Taken together these observations suggest that CNX and FAM134B form a stable complex that does not require PC binding to occur. This interaction seems to occur within the ER membrane since it is mediated by the reticulon homology domain of FAM134B. Notably, the reticulon homology domains generate membrane curvature by increasing the area of the cytoplasmic leaflet. These observations suggest that the binding of PC to CNX may induce a conformational change of the reticulon homology domain that increases ER membrane curvature, favoring vesicle formation. We have discussed this possibility in the discussion section.

Additional comments

- Add quantification and statistical information to all images.

The quantification has been performed for the microscopy data in main figures (except for the ones in the new Fig. 2 as it is not considered suitable). The quantification for figures 1E and F are in Fig 3A (Saos2, mock) and 3B (MEFs, WT), quantification of RCS has been previously shown (Cinque, Forrester et al. *Nature* 2015). We have also reported in figure legends that "western blots are representative of three independent experiments". Statistical information has been added to all analyses performed.

- In pulldown experiments the input and the pulldown should be in the same gel.

In the pulldown experiments reported in figure 7C, the input and the pulldown are in the same gel (unnecessary lanes were removed). For more clarity, we changed the figure accordingly. We removed the pulldowns with HALO collagen in figure 7B and 7C, as they were showing redundant results with the pulldown experiments now shown in figure 7C, and we included additional co-immunoprecipitation experiments that better characterized the CNX-FAM134B interaction (Fig 7A,B and EV7A).

- Add Western blots with autophagic markers and quantifications to all experiments shown in figures 1-3

We added western blots with autophagy markers to the experiments shown in Fig. 1 and 3 (this new version does not contain previous data of Fig. 2 anymore. Data are now in Fig. 1J and Fig. EV 2A and 5A). These data clearly show that PCs accumulate similarly to the canonical autophagy substrates in cells with defective autophagy or lysosome function.

- WB with anti-FAM antibodies is missing from the experiment shown in figure 4b, moreover some of the blots (LC3, actin and VAPA) seem overexposed.

The blots (LC3, β -actin and VAPA) were changed for less exposed images (Fig. 4A and Fig. EV5B). Moreover, we performed a FAM134B blot in CRISPR FAM134B cells compared with wild type MEFs. The blots are presented in Fig. 4A and Fig. EV5B.

-The experiment shown in figure 5E is missing the analysis with anti-CNX and anti-LC3 antibodies.

We completed the blot as suggested by adding CNX and LC3 markers (now Fig. 5B).

-How the authors explain the LC3 band shown in HALO pulldown (figure 7b). WB with anti-HALO is missing (in both 7b and c). The data presented in figure 7c are not convincing. The changes in mobility CNX are not explained and the effect of CST is marginal.

As stated previously, we decided to remove the pulldowns with HALO collagen in figure 7B and 7C, as they were redundant with the immuno-precipitation in figure 7D. We therefore modified figure 7 accordingly.

Minor comment:

Specify the bafA treatment in the actual images may help.

We now specified all the experiments that were performed with BafA1 treatment in the actual images.

Referee #2:

In this manuscript, Forrester and colleagues report a role for lysosome and autophagy in the degradation of misfolded procollagen protein mediated by the ER-phagy receptor FAM134B and the ER protein CNX (calnexin). Overall, the results sound interesting and the topic fits more or less with the general interest for the cell and molecular biology community and the readers of EMBO Journal. However, the pivotal question concerning the central role of CNX (and CNX-FAM134B interplay) in broader specialized ER-phagy is not really addressed, despite the demonstration of its importance in pro-collagen degradation. Based on this, I am not totally convinced that this story must be published in EMBO journal as it is describing (pretty well) only the pro-collagen lysosomal degradation, but has no general impact on the field of ER/autophagy interplay.

MAJOR CONCERNS:

As written in the summary section, my principal remark concerns the importance/originality of the message, regarding the fact that FAM134B-mediated ER-phagy has been nicely studied and documented previously (including by authors of the present study) and that the relation between procollagen and lysosomal degradation has been suggested already (see review by same authors, in Matrix Biol, June 2018) but without questioning the relevance of this system for other cargoes. Indeed, as authors conclude that "the [CNX-FAM134B complex] is responsible for a specific ER-clearance mechanism that is crucial for ER and cellular homeostasis", this would be probably relevant to decipher whether or not this "FAM134B-CNX" ER pathway is indeed dedicated to a broad (or not) range of unfolded proteins turnover and to analyze this could be associated with stress situations in specific context(s) (such as bone physiology) and related to ER stress sensing.

We thank the referee for these important comments. To understand whether the CNX-FAM134B complex is selective towards collagens or if it is also involved in the degradation of other ER clients, we performed proteomics in FAM134B^{-/-} and CNX^{-/-} MEFs and found that only 17 proteins significantly accumulated in both FAM134B and CNX deficient MEFs compared with control cells. Strikingly, collagens (including Col1a1, Col1a2, Col6a1, Col6a2, Col5a1) and collagen-interacting proteins (Procollagen C-endopeptidase enhancer 1, SPARC/Osteonectin) were predominantly present in this list. These data strongly suggest that the CNX-FAM134B complex selectively delivers procollagens to the autophagy/lysosome pathway (Fig. 6 A,B,C and EV Fig. 6). Thus we have identified a new example of selective autophagy devoted to the removal of misfolded procollagens from the ER. Collagens are the most abundant proteins of the animal kingdom and the main component of tissue extracellular matrices. Furthermore, defects in collagen proteostasis are associated with several human diseases. Hence the identification of CNX-FAM134B-mediated ER-phagy as a collagen ER-quality control mechanism not only represents one of the first examples of

how cross-talk between ER quality control machinery and ER-phagy exerts cargo selectivity, but it may also have future therapeutic implications.

This pathway is very likely to be physiologically relevant, particularly during bone growth. Indeed, we have recently reported that the inhibition of autophagy or lysosome function in chondrocytes leads to the accumulation of PC2 molecules in the ER of chondrocytes, leading to ER enlargement and impaired function. As a result, mice with defects in the lysosome and autophagy pathways have bones with lower amounts of collagens and develop post-natal bone growth retardation (Cinque, Forrester et al. *Nature* 2015, Bartolomeo et al. *JCI* 2017).

A recently published paper in EMBO journal from the group of Dr. Molinari demonstrated that the CNX-FAM134B complex also mediates the delivery of proteasome-resistant, ER resident, polymers of alpha1-antitrypsin Z (ATZ) to the lysosomes for degradation (Fregno et al. doi: 10.15252/emj.201899259). This pathway was named ERLAD and has substantial differences compared to the autophagy of collagen reported here:

- 1) ERLAD mediates the clearance of mutant ATZ, suggesting that it can be activated under particularly stressful conditions (e.g. presence of large ER aggregates). Conversely, autophagy of collagen is a quality control process, required in collagen producing cells even in the absence of PC mutations.
- 2) ERLAD does not require the formation of autophagosomes since ATZ is delivered to lysosomes via ER-derived vesicles. Consistently, several autophagy proteins are dispensable (e.g. ATG9, ULK1, FIP200, ATG13) for ERLAD to occur. Conversely, procollagen molecules are delivered to lysosomes by autophagosomes whose biogenesis requires the activity of the autophagy proteins.
- 3) ERLAD requires CNX but not the activity of the other enzymes and chaperones of the calreticulin/calnexin cycle, suggesting a yet to be discovered atypical role of CNX. Conversely, PC molecules destined to FAM134B-mediated ER-phagy are substrates of the calreticulin/calnexin cycle.

All in all these data suggest that the CNX-FAM134B complex is implicated in two distinct, but functionally related clearance pathways. We have discussed these aspects in the "discussion section".

The conclusions presented in the current manuscript are most of the time well supported by experimental data, but the dynamic aspect of the described pathway is lacking: notably, the correlative light electron microscopy used in the paper (Figure 6) is not very informative since it described only one situation in time. It would have been more useful to assess the pathway dynamically, by videomicroscopy for example, to document the capture of PC-positive ER structure by pre-existing autophagosome and then its delivery to the lysosome. Moreover, there is no precise information about autophagic machinery recruitment at CNX specific zones of the ER engaged with PC handling. Are these domains positive for PI3P, DCFP1, ULK1, etc?

Following the reviewer's suggestion we performed videomicroscopy to assess the pathway dynamically. Specifically, we generated two cell lines: 1) U2OS cells co-expressing GFP-LC3, mcherry-PC2 and RDEL-HALO (ER marker); 2) U2OS expressing GFP-2-FYVE (PI3P binding probe), mcherry-PC2, and RDEL-HALO.

Videos and still images clearly show the sequestration of PC2-positive ER structures by GFP-2-FYVE -and GFP-LC3 structures (Fig. 2A,B and Movies EV1 and 2).

Furthermore, by super-resolution confocal microscopy (Airyscan) we show DFCP1- and LC3-puncta that co-localize with CNX-positive PC1 molecules (FIG. 2C and EV3A). These new data, together with the CLEM analysis and the data obtained using KO cells, clearly support the model by which PC molecules are sequestered in CNX positive ER regions that are then subjected to autophagy.

Finally, there is not real data on the mechanisms by which misfolded PC can be sequestered together with enriched CNX in a given subdomain of the ER: such data could highlight a dynamic process by which chaperones such as CNX could trigger spatial sorting in the ER membrane, to facilitate ER piecemeal autophagy process.

To understand whether CNX could trigger spatial sorting in the ER membrane and facilitate ER piecemeal and autophagy processes we analyzed the sequestration of FAM134B in LC3 positive vesicles in WT and CNX ^{-/-} MEFs. We found no significant change in the FAM134B-LC3 co-localization extent between WT and CNX ^{-/-} MEFs, suggesting that the sequestration of FAM134B-positive ER structures into autophagosomes does not require the activity of CNX (Fig. EV 7B,C).

These observations together with the fact that PC molecules do not efficiently reach lysosomes in CNX ^{-/-} MEFs or in WT MEFs treated with castanospermine suggest that CNX is rather involved in the selective recognition of misfolded PC molecules destined for the lysosomes via autophagy. Calnexin is part of the calnexin/calreticulin cycle. This dedicated system at the ER assists folding and ensures the quality of final products before ER release. It requires components like calnexin and calreticulin (CRT), along with their associated co-chaperone ERp57 and folding sensor udi-glucose:glycoprotein glucosyltransferase (UGT1). In the revised version of the manuscript we have included new data showing that PC1 delivery to lysosomes requires ERp57, CRT and UGT1 proteins. These data suggest that the CNX/CRT quality control machinery selects PC molecules destined to degradation (Fig. 5C).

In summary (and to my opinion), this manuscript is closer of a "cell biology case report" concerning the degradation of PC by autophagolysosome (and again, most of the experiments are well addressed) than a broad cell biology message on ER-phagy and ER dynamics interplay in response to general misfolded proteins.

We believe that our study, together with Fregno et al. EMBO J. 2018, represent the earliest examples of how ER luminal cargoes can be disposed of from the ER and degraded by the lysosomes. The identification of CNX-FAM134B interplay explains the mechanism by which ER-phagy can be selective toward specific ER clients. This is a novel example of a crosstalk between the luminal ER-quality control machinery and a cytosolic degradative pathway. It is likely that future studies will identify other clients recognized by FAM134B-mediated ER-phagy through the interaction of FAM134B with different ER chaperones.

Furthermore, it is important to emphasize that our work describes a new pathway specific for the control of collagen proteostasis. Indeed, our new mass spectrometry analyses clearly show that this machinery is devoted to the degradation of different types of collagens, suggesting that cells may have evolved a specific mechanism to cope with the difficulties associated with the production and secretion of procollagens in the ER. This is not surprising if we consider that collagens are the most abundant proteins of our body (about 25% of our dry weight), and that its production represents a major task for cells. Consistently, defects in collagen folding and secretion are associated with several disease conditions.

MINOR POINTS:

- 1) Regularly, authors refer to "a fraction" of PC that is misfolded, but readers lack details to understand the physiological relevance of such amount of misfolded PC: is that cell type dependant? Could it be modulated by given physiological or pathophysiological situations, including stress and autophagy inducing treatments? Could be related somehow to autophagy defects or modulating situations?***

We quantified the amount of intracellular collagen found in autophagosomes relative to the total amount of intracellular collagen in MEFs, Saos2 and RCS cells. As shown in the graph below, results were similar among the different cells, although RCS showed a slightly higher value.

We also investigated whether autophagy modulation could induce the clearance of an excess of wild type (WT) and mutant collagen molecules from the ER. We studied two missense mutations (R789C and G1152D) in the Col2a1 protein that induce misfolding of the PC2 triple helix and accumulation within the ER of chondrocytes. The mutations cause a type II collagenopathy in humans, named Spondyloepiphyseal Dysplasia Congenita (SEDC) (Murray 1989). When expressed in chondrocytes the R789C- and G1152D- mutants were targeted to the lysosomes at higher rates compared with WT PC2. Notably, pharmacological enhancement of autophagy with the autophagy inducing peptide Tat-Beclin-1 increased targeting of WT and (to a lesser extent) of mutant PC2 molecules to lysosomes. Opposite results were observed by treating cells with the autophagy inhibitor SAR405. These data have been included in the manuscript (Fig. EV4).

Taken together, these data suggest that the fraction of PC delivered to lysosomes via autophagy depends on cell type, on procollagen folding efficiency and on autophagy levels.

2) *the experimental rationale of data presented in Figure 1 G-H is not very clear and should better explained.*

We modified the text as follows: “Lysosomal storage disorders (LSD) are genetic diseases characterized by a defective lysosomal degradative capacity due to mutations in genes encoding for lysosomal proteins. As a result, lysosomal substrates progressively accumulate within the lumen of lysosomes causing lysosomal swelling and cell dysfunction. We sought to determine whether PC molecules accumulate in the lysosomes of LSD osteoblasts. Saos2 osteoblasts in which the alpha-L-iduronidase gene was deleted using CRISPR-Cas9 technology (CRISPR-IDUA) represent a disease model of mucopolysaccharidosis type I (MPS I), a lysosomal storage disorder with severe skeletal manifestations (Oestreich AK, 2015). Similarly to cells treated with BafA1, CRISPR-IDUA showed swollen lysosomes, suggesting an accumulation of undigested substrates in the lysosomal lumen (Fig 1I). Most importantly, the level of PC1 in lysosomes, and in the whole cell lysate, was higher in CRISPR-IDUA Saos2 compared to control cells (Fig 1I,J).”

3) *the use of ATG9 CRISPR cell line is not clear to me: why ATG9? Usually, studies aimed at investigating implication of autophagy (or autophagic machinery) in a given process are often addressed with genes directly implicated with autophagosome biogenesis (including genes from class III PI3K such as Beclin1, ATG14 or VPS34) or LC3 lipidation regulation (such as ATG7, ATG5, ATG12 or ATG16). It is not clear why data with ATG-/- MEFs are presented independently from ATG9 data... The putative role of ATG9 in non autophagic or non-canonical autophagy related processes is not very in favour of the conclusion of authors claiming that “.. autophagy is responsible for*

delivery of PC molecules to lysosomes for degradation". Manuscript/figures could be modified to make it clearer.

We removed ATG9 data since they were redundant with the other data obtained using MEF KO for Fip200, Atg7 and Atg16L and Saos2 silenced for several autophagy genes.

- 4) *the discussion section could benefit of more hypotheses concerning a putative new function for ER chaperones in selective sorting of misfolded proteins for autophagy / ER-phagy.*

The discussion section has been substantially improved. The new parts have been highlighted in yellow throughout the section.

- 5) *authors refer to LAMP1 as a lysosomal marker, while the precise localization of LAMP1 is lysosome and late endosomes.*

We corrected the text as follows: "PC molecules progressively accumulated in the lumen of swollen endo/lysosomes (Lamp1 positive organelles, hereafter referred as lysosomes)"

Referee #3:

In my view, this is an outstanding paper that uses many approaches to disclose the mechanism of autophagy quality control of procollagen (PC) via a Calnexin-FAM134B ER-Phagy complex. The principal conclusions are that canonical autophagy mediates PC delivery from ER to lysosomes; that the ER-resident autophagy receptor FAM134B is a key regulator in the process; that Calnexin (CXN) interacts with FAM134B and they in turn interact with LC3 to form a novel ER-phagy complex which provides the clearance mechanism for delivery of PC from ER to lysosomes for turnover.

The paper substantially advances the field. It is very well written and is a pleasure to read. Experiments are well designed and data presented clearly. The combination of biochemical, cell biological and genetic techniques is quite impressive and, in toto, provide definitive evidence to support the conclusions.

My one minor suggestion is to change the legend to Figure EV4 from a summary statement to one that describes what is shown in the Figure. Eg, Suggest it read " Figure EV4 Unlike with PC1 (Fig 5C,D), CNX deficiency fails to increase accumulation of other resident ER Proteins "

We thank the referee for his/her positive comments. We have changed the legend of figure EV4 according to his suggestion.

Thank you for submitting a revised version of your manuscript. It has now been seen by two of the original referees and we have received their comments, which are enclosed below for your information.

As you can see, they both find that all criticisms have been sufficiently addressed and recommend the manuscript for publication.

REFeree REPORTS

Referee #1:

The authors successfully addressed the comments from the first round of review and in its present form the manuscript meets EMBO J scientific merit.

Referee #2:

The revised version of the "Forrester et al" manuscript on the calnexin selective autophagy pathway has considerably increased the quality and the clarity of the message. The authors have made a real and successful attempt to deal with reviewer 1 and my own comments; especially, the mass spectrometry data and the mechanistic description of local autophagic events at ER membrane experiments are of top-quality, and authors should be commended for their efforts.

Corresponding Author Name: CARMINE SETTEMBRE

Journal Submitted to: The EMBO Journal

Manuscript Number: EMBOJ-2018-99847